# Constraining CMIP6 sea ice simulations with ICESat-2

Alek Petty<sup>1</sup>, Christopher Cardinale<sup>1</sup>, Madison Smith<sup>2</sup>

<sup>1</sup>Earth System Science Interdisciplinary Center (ESSIC), University of Maryland, College Park, MD, USA

<sup>2</sup>Woods Hole Oceanographic Institution (WHOI), Falmouth, MA, USA

Correspondence: Alek Petty (akpetty@umd.edu)

#### Abstract

This study evaluates sea ice simulations from the Coupled Model Intercomparison Project Phase 6 (CMIP6) using modern-era satellite measurements of sea ice area, total freeboard, and thickness. Current global climate models (exhibit substantial uncertainties in simulating sea ice, with significant contributions from both model uncertainty and internal variability. In our study, simulated Arctic and Southern Ocean total freeboard and Arctic winter sea ice thickness are assessed with data from NASA's Ice, Cloud, and land Elevation Satellite-2 (ICESat-2) mission, providing an additional constraint beyond traditional passive-microwave sea ice area comparisons used extensively in previous studies. Freeboard comparisons benefit from accurate observations from satellite laser altimetry but motivate increased focus on bulk sea ice density estimates across models and observations. The short observational time period also increases the role of internal variability. We undertake a plausibility assessment where we account for both observational uncertainty and internal variability across our different metrics for both hemispheres. In general, we see more plausible metrics in the Arctic compared to the Southern Ocean, with important differences when analyzing annual means vs. March and September means. We provide an example of this plausibility assessment by producing constrained estimates of 2015-2035 seasonal sea ice volume, using model subsets constrained using either area metrics or the combined area, freeboard and thickness metrics, with freeboard and thickness providing important additional impacts in terms of the mean seasonal cycle and spread. Finally, we present regional comparisons and a composite analysis, with models showing systematic underestimation of thicker ice in the western Arctic and clear differences in the simulation of Eastern/Western Arctic sea ice. Overall, our study provides novel insights into sea ice model evaluation and emphasizes the potential benefits of integrating altimetry data from ICESat-2, as well as providing a discussion on the potential utility of these model constraints and future research priorities.

Deleted: GCMs)

Deleted: ICESat-2

## 1 Introduction

Earth's polar sea ice cover is undergoing rapid declines in response to anthropogenic climate change (Intergovernmental Panel on Climate Change, 2023). Coupled global climate models (GCMs) are commonly used to simulate past and future sea ice conditions and disentangle the associated impacts and feedbacks with the rest of the climate system (Notz & SIMIP Community, 2020; Goosse et al., 2018; Jahn et al., 2024; Pithan and Mauritsen, 2014; Smith et al., 2019). Historical GCM outputs are also used to provide important constraints on sea ice mass, energy and freshwater budgets (Holland et al., 2006, 2010; Keen et al., 2021; Massonnet et al., 2018; Meredith et al., 2019; Zanowski et al., 2021) and can provide training input for seasonal/sub-seasonal forecasting models (Andersson et al., 2021). These efforts are often hindered by the large and poorly constrained uncertainties of the sea ice state in current GCMs.

Uncertainty in GCM sea ice conditions arise from the combined impact of model structural uncertainty and internal variability, with additional contributions of forcing uncertainty in future scenario runs. Model uncertainty is typically estimated based on the spread across available models. Model uncertainty has numerous causes, including biases in atmospheric/ocean forcing and errors in model physics (Massonnet et al., 2018). In GCMs, sea ice has historically been considered a simple boundary condition that increases the surface albedo and alters the surface energy balance, rather than being a crucial climate component in and of itself. Many of the models included in Coupled Model Intercomparison Project 5 (CMIP5) feature only basic parameterizations of sea ice, with only a few models including significant improvements to the underlying sea ice physics schemes in the newly released CMIP6 suite (Notz & SIMIP Community, 2020, refered to as SIMIP2020). However, improvements in sea ice simulation in CMIP6 have been suggested, alluding to improvements in polar atmospheric/ocean forcing and/or model physics (Notz & SIMIP Community, 2020; Roach et al., 2020; Davy and Outten, 2020). Internal variability, which represents the random fluctuations of the climate system, can provide a significant and irreducible source of additional sea ice state uncertainty. The significant uncertainty across CMIP6 sea ice simulations poses important questions about its potential utility for predicting future sea ice conditions, e.g. the potential timing of an ice-free Arctic (Jahn et al., 2024).

There is no agreed upon approach for analysing multi-model sea ice ensembles. In the simplest approach of full model democracy, all models are considered equally plausible, no exclusion or calibration is employed, and the model uncertainty remains unchanged. This is often the approach taken when observations are too unreliable to provide sufficient constraint, or if internal variability estimates are unavailable. Beyond this, model runs can be excluded or weighted based on assessments of the combined observational uncertainty and internal variability. The exclusion approach was adopted in SIMIP2020, whereby models with historical Arctic sea ice area significantly outside a plausible range calculated from a combination of observational uncertainty and internal variability were omitted from the final CMIP6 future Arctic sea ice projection analysis. More sophisticated methods for excluding and/or weighting models based on comparisons with observations are available, including methods to recalibrate the models based on their simulated sea ice response to temperature variability (Bonan et al., 2021),

Deleted: has

Deleted: suggesting

atmospheric circulation (Topál and Ding, 2023) and greenhouse gas forcing (Kim et al., 2023). Similarly, emergent constraints, an approach that utilizes statistical relationships between observable quantities and future model diagnostics, has been used to constrain sea ice projections using a variety of metrics (Massonnet et al., 2018; Thackeray and Hall, 2019; Wang et al., 2021). These methods all generally rely on the assumption that model performance is consistent across time periods and depend crucially on the specific research question posed (Notz, 2015). However, recalibration approaches generally do not update associated state variables (e.g., impact on the surface energy budget or freshwater fluxes from the sea ice to the ocean), and can be more challenging to implement, motivating continued attention on optimal exclusion or weighting approaches.

Figure 1: Schematic showing (left) observations of sea ice total freeboard from laser altimetry towards estimates of sea ice thickness, (middle) advanced physics and ice thickness distribution typical of state-of-the-art sea ice models, (night) fixed ice thickness/snow parameterization used in more basic sea ice models. Note that ice freeboard is the extension of sea ice above sea level, while total freeboard is the extension of the ice and snow layer above freeboard. Sea ice models do not typically simulate freeboard, but this can be estimated based on an assumption of isostacy.

To-date, most of the sea ice model exclusion and calibration efforts have utilized observational estimates of sea ice area or extent from the long-term (>40 year) passive microwave record (Lavergne et al., 2019; Parkinson, 2019). Passive microwave sensors measure brightness temperature at different frequencies and polarities and use this information to infer the concentration of sea ice over the ocean surface, which can then be converted to sea ice areal coverage (by multiplying by the grid-cell area) and extent (grid-cells with at least 15% sea ice concentration). The multi-decadal time-period benefits from being long enough to be representative of long-term climate conditions and reduce the role of internal variability. It can also be used to assess the sensitivity of sea ice area to warming or carbon dioxide forcing as an additional observational constraint. Sea ice area uncertainty is often estimated by comparing across different observational products (Notz & SIMIP Community, 2020) or by assuming fixed values/percentages based on community consensus (Massonnet et al., 2012). However, sea ice concentration provides only limited information within the more consolidated ice pack, where large gradients in thickness/volume manifest (Petty et al., 2023).

Dedicated polar-focussed satellite altimetry missions have been launched since the early 2000s that can accurately profile sea ice height towards estimates of sea ice freeboard, thickness and, thus, volume. These include the National Aeronautics and Space Administration (NASA) Ice, Cloud, and land Elevation Satellite (ICESat) mission (2003-2009, Zwally et al., 2002;

Deleted: CO2

Deleted: and

Kwok and Cunningham, 2008), the European Space Agency (ESA) CryoSat-2 radar altimetry mission (2010 onwards, Laxon et al., 2013) and, most recently, NASA's ICESat-2 laser altimetry mission (2018 onwards, Neumann et al., 2019; Petty et al., 2020). The altimeters measure the height of sea ice and open water leads between ice floes. The open water height estimates are used to generate an estimate of the local sea surface height. Differencing the local sea surface height from the sea ice height provides an estimate of freeboard, the extension of sea ice above sea level. See basic schematic in Fig. 1. Laser altimeters (e.g., NASA's ICESat and ICESat-2) provide estimates of the snow-covered ice surface height (a metric commonly referred to as total freeboard). In contrast, data from radar altimeters (e.g. ESA's CryoSat-2) are typically used to provide an estimate of the less distinct ice-snow interface height and thus an estimate of ice freeboard. The effective radar penetration depth at Ku-band used in CryoSat-2 is generally considered to come from the ice-snow interface, although studies continue to challenge this (Nandan et al., 2017; Willatt et al., 2011). The satellite laser altimeters benefit from higher spatial resolution on the ground than radar (footprints of 10s of meters as opposed to hundreds of meters to kilometres). They also profile the upper snow surface thus providing a useful constraint on total snow loading. However, radar is unaffected by clouds and CryoSat-2 benefits from continuous data collection and refinement since it launched in 2010. Additional input assumptions regarding snow depth, snow density and ice density together with an assumption of isostasy are then typically used to convert measured freeboard (ice or total depending on the sensor) to an estimate of sea ice thickness for the Arctic, which introduces significant additional uncertainty to this 'observational' estimate (Giles et al., 2007; Kwok and Cunningham, 2008; Petty et al., 2020). Constraining the thickness uncertainty remains challenging due in-part to the lack of ground-truth data available for validation. For Antarctic sea ice, limited knowledge of the more complex snow loading has generally hindered production of similar snow and sea ice thickness data production efforts to-date, although novel approaches show promise (Fons et al., 2021; Garnier et al., 2021).

105

Due to concerns around accuracy and uncertainty quantification, combined with their more limited temporal coverage, sea ice thickness data have generally been excluded from model assessment efforts to-date (Notz & SIMIP Community, 2020; Kay et al., 2022; Roach et al., 2020), despite the fact that the mean thickness has been demonstrated to be the crucial factor controlling sea ice variability and trends (Massonnet et al., 2018). With the recent ICESat-2 period now extending into its seventh year of successful data collection (at the time of writing) and the improved understanding achieved from the joint operation of both 120 ICESat-2 and CryoSat-2, we can begin to reconsider the concerns around accuracy and time-period. In addition, comparisons of the direct observations of total freeboard with model estimates of this same quantity may offer another path forward for model assessments, better leveraging the high accuracy of the ICESat-2 laser altimeter observations. Assessments of total freeboard avoid a significant component of the thickness uncertainty that is introduced in the conversion of freeboard to thickness. Laser altimetry total freeboard estimates also avoid the uncertainties associated with radar freeboard profiling (e.g. 125 identifying which interface is dominating the radar return). Total freeboard comparisons are expected to provide significant value in the Southern Ocean, where snow depth and sea ice thickness estimates are less reliable. Southern Ocean sea ice is also thought to be composed of a higher fraction of thinner first-year ice with higher snow depths compared to the Arctic, such that total freeboard could be considered a better proxy for thickness than in the Arctic (Kurtz and Markus, 2012; Worby et al.,

Deleted: and

2008). In addition, total freeboard is utilized in some sea ice model parameterizations, e.g. atmospheric form drag (Tsamados et al., 2014) and snow-ice formation (Hunke et al., 2015), providing an additional motivation to assess its representation in models, especially as we prepare for the upcoming release of CMIP7 output. However, total freeboard integrates information from both ice thickness and snow depth variability concurrently, meaning changes in freeboard can be linked to changes in the underlying ice thickness and/or snow depth and diagnosing the cause of total freeboard biases is more challenging than the more direct prognostic variables of area and thickness. Additionally, freeboard is not a prognostic variable in sea ice models, and so it is typically calculated within the model as needed assuming hydrostatic equilibrium from the associated ice state variables (see schematic in Fig. 1 and Eq. 2 in Sect. 2.1 below), either in a post-processing step or within the relevant model parameterization scheme.

Following a request from the Sea Ice Model Intercomparison Project (SIMIP) in the lead up to CMIP6 (Notz et al., 2016), several modelling groups provided direct outputs of ice freeboard, making a comparison effort timely. To our knowledge, we are unaware of any studies that have explored this new model output. In this paper, we thus undertake a first attempt at using satellite altimetry observations of total freeboard from NASA's ICESat-2 mission to evaluate sea ice output from CMIP6. We compare these with evaluations using the traditional sea ice area and arguably more uncertain (winter Arctic) sea ice thickness metrics derived from satellite observations in order to demonstrate the advantages and disadvantages of these comparisons and look ahead to using this information to better constrain simulations and projections of sea ice across both poles. Producing a longer-term record of sea ice area, freeboard and thickness through assimilation of sea ice altimetry data into a consistent reanalysis is a current focus of the sea ice community, although challenges unique to sea ice are only recently being assessed (Riedel and Anderson, 2023; Wieringa et al., 2023; Williams et al., 2023). We thus focus only on direct ICESat-2 observations

## 150 2 Data and Methods

#### 2.1 CMIP6 sea ice model output

We use sea ice model output from the CMIP6 archive (Eyring et al., 2016). CMIP6 data are officially hosted through the Earth System Grid Federation (ESGF), enabling users to directly download all relevant CMIP6 output made available by all contributing model centres using the Open-source Project for a Network Data Access Protocol (OPeNDAP) system. We additionally make use of the Pangeo Analysis-Ready Cloud-Optimized CMIP6 catalogue (https://pangeo-data.github.io/pangeo-cmip6-cloud/) which is hosted on both Amazon Web Services (AWS) and Google Cloud Project (GCP) cloud storage services, simplifying the data ingest process (where we use the NASA-funded AWS hosted CryoCloud platform, https://book.cryointhecloud.com and thus the AWS-hosted catalogue). Some model outputs are missing from the Pangeo cloud catalogues, so we utilize the ESGE/OPeNDAP data where needed to ensure full model selection. We primarily use data from the SSP2-4.5 future scenario (2015-2100) to ensure full overlap with our ICESat-2 observational period (2018 to 2024). The

in this study, while recognizing the challenges in characterizing internal variability from this short time-period.

Deleted: Section

**Deleted:** is

Deleted: P

Code Availability section provides links to all code used to retrieve and analyse these data. We use the monthly mean output

across all models except for AWI-CM-1-1-MR which only provides daily data which we average to monthly.

As our study builds on the initial SIMIP CMIP6 sea ice evaluation studies which only included models available in the initial IPCC AR6 analysis time frame (Notz & SIMIP Community, 2020; Roach et al., 2020), we briefly explored differences in Arctic sea ice area from the model runs used in our study compared to the SIMIP2020 study. Overall, our study uses 13 more models compared to the SIMIP2020 study. Despite the significant difference in model subsets, differences in the multi-model CMIP6 mean Arctic sea ice area over a 2015-2035 time-period in both March and September were negligible (see

Table 1: Model variables used in our study, its denoted symbol if used in the derivations below, the official CMIP6 variable name, and the units. CMIP6 variables denoted N/A are not provided directly and are either prescribed or calculated in this study.

Supplementary Information Fig. S1).

| Variable name                | Symbol     | CMIP6<br>variable | <u>Units</u>       |
|------------------------------|------------|-------------------|--------------------|
| Sea ice area                 | <u>N/A</u> | siconc            | <u>m</u>           |
| Sea ice thickness            | $h_i$      | sithick           | m                  |
| Sea ice freeboard            | $h_{fi}$   | <u>sifb</u>       | <u>m</u>           |
| Total freeboard              | $h_{ft}$   | <u>N/A</u>        | <u>m</u>           |
| Snow thickness               | h          | sisnthick         | m                  |
| Sea ice mass per unit area   | $M_i$      | simass            | kg                 |
| Sea ice volume per unit area | Vi         | sivol             | m                  |
| Bulk sea ice density         | Pi         | <u>N/A</u>        | kg m <sup>-3</sup> |
| Bulk snow density            | PA         | <u>N/A</u>        | kg m <sup>-3</sup> |
| Seawater density             | Prox       | <u>N/A</u>        | kg m <sup>-3</sup> |

The grid-cell mean sea ice variables used in our study include the following (CMIP6 variable names in parentheses): sea ice area (siconc), sea ice thickness (sithick), sea ice freeboard (sifb), snow thickness (sisnthick), sea ice mass (simass) and sea ice volume (sivol), as summarized in Table 1. Note that the grid-cell mean ice freeboard variable was requested from CMIP6 contributing centres by the Sea-Ice Model Intercomparison Project (SIMIP) consortium at priority level 2 (Notz et al., 2016) and only 17 of the modelling centres provided this output for the SSP2-4.5 scenario runs. A listing of the CMIP6 models and the relevant variable availability is shown in Table 2. For the models that do not provide ice freeboard output we can instead estimate this from the provided variables of ice and snow thickness, estimates of ice and snow density, and an assumption of

Deleted: and ... SIMIP Community, 2020; Roach et al., 2020), we briefly explored differences in Arctic sea ice area from the model runs used in our study compared to the SIMIP2020 study. Overall, ...ore models compared to the SIMIP2020 our study uses 11 study. The SIMIP2020 study additionally uses the AWI-CM-1-1-MR data, but this model was not included in our study to ensure consistency across the models as only daily data were available in our search (with all our analysis based on monthly mean outp ... [1] Formatted: Caption, Space After: 6 pt, Keep with next Formatted: Centered, Right: -0.09 Formatted: Centered Formatted: Centered **Formatted Table** Formatted: Font: 9 pt Formatted: Font: 9 pt Formatted (... [2] Formatted: Font: 9 pt Formatted: Font: 9 pt **Formatted** (... [3] Formatted: Font: 9 pt Formatted: Font: 9 pt Formatted (... [4] Formatted (... [5]) Formatted: Font: 9 pt Formatted: Font: 9 pt Formatted (... [6]) Formatted: Font: 9 pt Formatted: Font: 9 pt Formatted (...[7]) Formatted: Font: 9 pt Formatted: Font: 9 pt **Formatted** (... [8] Formatted: Font: 9 pt **Formatted** (... [9]) **Formatted** (... [10]) Formatted: Font: 9 pt

**Deleted:** . . . . ote that the grid-cell mean ice freeboard vari (... [13])

(... [11])

(... [12])

Formatted: English (US)

Formatted

**Formatted** 

Formatted: Font: 9 pt

isostacy. This method can also be used to check consistency with the provided ice freeboard output from the models that do provide that output. Starting with the hydrostatic equilibrium equation for ice thickness:

$$h_i = \frac{h_{fi}\rho_\omega + h_s\rho_s}{(\rho_\omega - \rho_i)},\tag{1}$$

where  $h_{fi}$  is sea ice freeboard,  $h_s$  is snow thickness,  $\rho_{\omega}$  is seawater density (1024 kg m<sup>3</sup>),  $\rho_s$  is the bulk snow density, and  $\rho_i$  is the bulk ice density. We can rearrange Eq. 1 to calculate ice freeboard as:

$$h_{fi} = \frac{h_i(\rho_\omega - \rho_i) - h_s \rho_s}{\rho_\omega}.$$
 (2)

This ice freeboard can be converted to an estimate of total freeboard by simply adding snow thickness (sisnthick) as:

$$h_{ft} = h_{fi} + h_s \,. \tag{3}$$

Snow thickness was listed as a priority level 1 variable in (Notz et al., 2016) and is provided by the 36 models that provide either the grid-cell sea ice thickness or sea ice volume and area. For the models that do provide outputs of ice freeboard, we generally assume this is calculated in post processing as in Eq. 2. It is worth noting that differences could arise both from the calculation of freeboard at sub-monthly time-steps before averaging to monthly, as well as from using the categories of ice thickness across the Ice Thickness Distribution (ITD) before averaging across the grid-cell (for the models that simulate an ITD).

A crucial additional variable in the conversion between freeboard and thickness is the bulk ice density ( $\rho_t$ ): the higher the ice and snow density, the lower the freeboard. Neither the bulk ice density nor snow density is provided directly by any of the CMIP6 groups, as generally it is not considered a prognostic variable and instead a prescribed constant. However, for some of the more sophisticated sea ice models, the effective bulk ice density can be considered a function of the variable internal temperature and salinity which varies based on the internal sea ice physics scheme and needs to be calculated during the ice freeboard calculation (e.g. CESM2, D Bailey, personal communication). To our knowledge, all CMIP6 sea ice models currently use a constant snow density of 330 kg m<sup>-3</sup><sub>x</sub> (this was also assumed in the CMIP6 sea ice freshwater analysis in Zanowski et al., 2021). Prescribed (or variable) bulk sea ice densities across CMIP6 was harder to determine from the available documentation. Alternatively, there are two ways in which we can infer the bulk ice density. The first is to infer bulk ice density from provided outputs of total (grid-cell mean) sea ice mass ( $M_t$ ) and volume ( $V_t$ ) from the 24 models that provide these outputs (listed in Table 2) as:

$$\rho_i = M_i / V_i \ . \tag{4}$$

We can also indirectly infer bulk ice density from the 17 models (listed in Table 2) that provide outputs of ice freeboard, ice thickness and snow thickness together with an estimate of snow density and seawater density through rearranging Eq. 2 in terms of bulk ice density as:

Deleted: ice
Deleted: /m³

Formatted: Superscript

Deleted: kg/m<sup>3</sup>

Deleted: Table

Deleted: 2

Deleted: 6

$$\rho_i = \rho_\omega - \frac{\rho_\omega h_{fi} + \rho_{Sh_S}}{h_i}. \tag{5}$$

We set the snow density to 330 kg m<sup>-3</sup> and the seawater density to 1024 kg m<sup>-3</sup> based on our review of the default options used across CMIP6 sea ice models.

Table 2: CMIP6 sea ice model summary and data variable availability. Models are only added if they provide relevant SSP2-4.5 output. Variables are only listed if used in this study. All data output is monthly and available on the native model grid. There are several exceptions: sicone is daily for AWI-CM-1-1-MR and data is only available on grIgrids (regridded) for INM-CM4-8, INM-CM5-0, KIOST-ESM. The first 17 rows indicate the freeboard output subset. Note that in most cases the underlying sea ice model has been adapted to ensure consistency with other model components. Semtner-Hibler refers to the Semtner zero-layer thermodynamics model
 (Semtner, 1976) and Hibler ice dynamics model (Hibler, 1979). Sea ice variables are all grid-cell monthly means and are summarized in Table 1. The final column describes whether any of the model output was used in the Notz & SIMIP Community (2020) study (S), the Roach et al., (2020) study (R), both (B), or neither (N). Sea ice model information from https://wcrp-cmip.github.io/CMIP6 CVs/docs/CMIP6 source id.html,

| <del>-</del>    |                |                                                 |            |     |
|-----------------|----------------|-------------------------------------------------|------------|-----|
| CMIP6 model id  | Sea ice model  | Variables                                       | S/R/B/N    | *   |
| ACCESS-CM2      | CICE5.1.2      | siconc, sithick, sifb, sisnthick, simass, sivol | В          |     |
| CESM2           | CICE5.1.2      | siconc, sithick, sifb, sisnthick, simass, sivol | В          |     |
| CESM2-WACCM     | CICE5.1.2      | siconc, sithick, sifb, sisnthick, simass, sivol | В          |     |
| CIESM           | CICE4          | sicone, sithick, sifb, sisnthick, simass, sivol | <u>N</u>   |     |
| CMCC-CM2-SR5    | CICE4.0        | siconc, sithick, sifb, sisnthick, simass, sivol | N          | - ✓ |
| CMCC-ESM2       | CICE4.0        | siconc, sithick, sifb, sisnthick, simass, sivol | N          |     |
| CNRM-CM6-1      | GELATO6.1      | sicone, sithick, sifb, sisnthick, simass, sivol | В          |     |
| CNRM-CM6-1-HR   | GELATO6.1      | sicone, sithick, sifb, sisnthick, sivol         | В          |     |
| CNRM-ESM2-1     | GELATO6.1      | sicone, sithick, sifb, sisnthick, simass, sivol | В          |     |
| HadGEM3-GC31-LL | CICE5.1.2      | sicone, sithick, sifb, sisnthick, simass, sivol | В          |     |
| IPSL-CM6A-LR    | LIM3           | sicone, sithick, sifb, sisnthick, simass, sivol | В          |     |
| MPI-ESM1-2-HR   | Semtner-Hibler | sicone, sithick, sifb, sisnthick, simass, sivol | В          |     |
| MPI-ESM1-2-LR   | Semtner-Hibler | sicone, sithick, sifb, sisnthick, simass, sivol | В          |     |
| MRI-ESM2.0      | MRI.COM4.4     | sicone, sithick, sifb, sisnthick, simass, sivol | В          |     |
| NorESM2-LM      | CICE5.1.2      | sicone, sithick, sifb, sisnthick, simass, sivol | <u>s</u> , |     |
| NorESM2-MM      | CICE5.1.2      | sicone, sithick, sifb, sisnthick, simass, sivol | N          |     |
| UKESM1.0-LL     | CICE5.1.2      | siconc, sithick, sifb, sisnthick, simass, sivol | В          |     |
| ACCESS-ESM1-5   | CICE4.1        | sicone, sithick, sisnthick, sivol               | В          |     |
| AWI-CM-1-1-MR   | FESOM1.4       | sicone, sithick, sisnthick, sivol <sub>▼</sub>  | В,         |     |
| BCC-CSM2-MR     | SIS2           | sicone, sithick, sisnthick, sivol               | В,         |     |
| CAMS-CSM1-0     | SIS1.0,        | sicone, sisnthick, sivol                        | В          |     |
| CanESM5         | LIM2           | sicone, sithick, sisnthick                      | В,         |     |
| CanESM5-1       | LIM2           | sicone, sithick, sisnthick                      | N.         |     |
| EC-Earth3       | LIM3           | sicone, sithick, sisnthick, sivol               | В          |     |
| EC-Earth3-CC    | LIM3           | sicone, sithick, sisnthick, sivol               | N          |     |
| EC-Earth3-HR    | LIM3           | sicone, sithick, sivol                          | <u>N</u>   |     |
| EC-Earth3-Veg   | LIM3           | sicone, sithick, sisnthick, sivol               | В          | •   |
|                 |                |                                                 |            |     |

Deleted: kg/m³

Deleted: 6

Deleted: kg/m³

Formatted: Left, Line spacing: single

Formatted: Font: 9 pt Formatted Table Formatted Table Deleted: B Deleted: S Deleted: siconc, sithick, sisnthick Deleted: BCC-CSM2-MR Deleted: SIS2 Deleted: siconc, sithick, sisnthick, simass, sivol Deleted: B Deleted: CAMS-CSM1-0 Deleted: SIS1.0 Deleted: siconc, sithick, sisnthick, simass, sivol Deleted: B Deleted: CanESM5 Deleted: LIM2 Deleted: siconc, sithick, signthick Deleted: B Deleted: CanESM5-1 Deleted: LIM2 Deleted: siconc, sithick, sisnthick Deleted: N Deleted: CIESM Deleted: CICE4 Deleted: siconc, sithick, sisnthick, simass, sivol Deleted: N Deleted: siconc, sithick, sisnthick, simass, sivol Deleted: siconc, sithick, sisnthick, simass, sivol Deleted: siconc, sithick, sisnthick, simass, sivol **Formatted Table** 

| EC-Earth3-Veg-LR | LIM3         | sicone, sithick, sisnthick        | N |
|------------------|--------------|-----------------------------------|---|
| FGOALS-f3-L      | CICE4.0      | sicone, sisnthick, sivol          | В |
| FIO-ESM-2-0      | CICE4.0      | sicone, sisnthick, sivol          | В |
| GFDL-CM4         | GFDL-SIM4p25 | sicone, sithick, sisnthick, sivol | В |
| GFDL-ESM4        | GFDL-SIM4p5  | sicone, sithick, sisnthick, sivol | В |
| KIOST-ESM        | GFDL-SIS     | sicone, sithick, sisnthick        | N |
| MIROC6           | COCE4.9      | sicone, sithick, sisnthick, sivol | В |
| MIROC-ES2H       | COCO4.9      | sicone, sithick, sisnthick        | N |
| MIROC-ES2L       | COCO4.9      | sicone, sithick, sisnthick        | В |
| NESM3            | CICE4.1      | sicone, sithick, sisnthick        | В |
| TaiESM1          | CICE4        | sicone, sithick, sisnthick, sivol | N |
| CanESM5-CanOE    | LIM2         | siconc                            | N |
| FGOALS-g3        | CICE4.0      | siconc                            | N |
| INM-CM4-8        | INM-ICE1     | siconc                            | В |
| INM-CM5-0        | INM-ICE1     | siconc                            | В |

#### 2.1.1 Model regridding

Model output was regridded to simplify analysis and enable spatial comparisons between the model and observations. We regrid all CMIP6 model data to rectilinear grids depending on the variable and observational comparison. For area we regrid all data to the EASE\_2.0 25 km x 25 km grid used by the concentration products described below, while for freeboard and thickness we use the North Polar Stereographic 25 km x 25 km grid used by the ATL20/IS2SITMOGR4 datasets, which are described in the following sections. We explored various options to optimize our regridding approach, utilizing the open-source Python xESMF package (https://xesmf.readthedocs.io/en/latest/index.html, Zhuang et al., 2024). We primarily utilized the conservative normed regridding method that preserves areal contributions of the input data within each observational grid-cell. To prevent unrealistic data interpolation along the coastline, land masks are specified for both the source and destination grids. For native model grids, land masks are calculated using the provided variable sea area percentage (sftof) which, when divided by 100, gives the fraction of the grid cell covered by ocean and us used in the conservative normed regridding described above. For the North Polar Stereographic 25 km x 25 km grid, the NSIDC land mask is used (Meier and Stewart, 2023). For the EASE2.0 25 km x 25 km grid, land is defined as NaN regions in the sea ice concentration data (this method is also used for models with no sftof data). We note that regridding can introduce artificial errors, but we ensured appropriate methods were used to minimize this and our investigations suggest negligible differences at both grid-cell and basin-scales (see Supplemental Information, Fig. S2-S3).

| Deleted: sicone, sithick, sisnthick                |
|----------------------------------------------------|
| Deleted: siconc, sithick, sisnthick                |
| Deleted: siconc, sithick, sisnthick                |
| Deleted: siconc, sithick, sisnthick, simass, sivol |
| Deleted: siconc, sithick, sisnthick, simass, sivol |
| Deleted: siconc, sithick, sisnthick                |
| Deleted: siconc, sithick, sisnthick, simass, sivol |
| Deleted: siconc, sithick, sisnthick                |
|                                                    |

Deleted: Table 1: CMIP6 sea ice model summary and data variable availability. Models are only added if found in both historical and SSP2-4.5 outputs. All data output is monthly and available on the native model grid. The first 16 rows indicate the freeboard output subset. Note that in most cases the underlying sea ice model has been adapted to ensure consistency with other model components. Semtner-Hibler refers to the Semtner zero-layer thermodynamics model (Semtner, 1976) and Hibler ice dynamics model (Hibler, 1979). Sea ice variables are all grid-cell monthly means and include: siconc = sea ice concentration, sithick = sea ice thickness, sifb = sea ice freeboard, sisnthick = snow thickness. The final column describes whether the model was used in the Notz and SIMIP Community (2020) study (S), the Roach et al., (2020) study (R), both (B), or neither (N). Sea ice model information from https://wcrp.

Deleted: OSI SAF

## 2.2 Observational sea ice data

## 2.2.1 Total freeboard and winter Arctic sea ice thickness from ICESat-2 altimetry

We use monthly gridded total freeboard data from NASA's ICESat-2 ATL20 product (Version 4) disseminated through the National Snow and Ice Data Center (NSIDC) (https://nsidc.org/data/atl20, Petty et al., 2023b). ATL20 is produced using a simple binning of the along-track freeboard data from the three strong beams of ICESat-2/ATLAS (ATL10, Kwok et al., 2023) on a 25 km x 25 km North Polar Stereographic grid. Significant data gaps can still exist for various reasons (e.g. cloud attenuation, lack of open water leads, spacecraft issues). Note that the underlying along-track ATL10 freeboard product masks data where sea ice concentration is less than 50% and where data are within 25 km of the nearest coastline. ATL20 data are available for both hemispheres across all months (November 2018 onwards). While data are available across the summertime Arctic, these data should be treated with more caution due to the lack of reliable melt pond classification scheme in the underlying sea ice processing (Tilling et al., 2020, Kwok et al., 2020). No uncertainty term is included in the products that account for the potential misclassification of leads as melt ponds, but the additional height filter in ATL10 (where only leads that pass a strict relative height filter are used to derive the local sea surface height) is expected to mitigate this issue to some degree.

In addition, we use monthly gridded winter Arctic sea ice thickness estimates from ICESat-2 (IS2SITMOGR4, Version 3) also disseminated through the NSIDC (https://nsidc.org/data/is2sitmogr4, Petty et al., 2023c). These data use snow loading estimates from the NASA Eulerian Snow On Sea Ice Model (NESOSIM, Petty et al., 2018), now at Version 1.1, a constant bulk ice density of 916 kg m<sup>3</sup>, and the isostacy assumption to derive estimates of sea ice thickness across the Arctic Ocean between September and April, since November 2018 (Petty et al., 2023c). The upgrade from IS2SITMOGR4 from Version 1 to 2 was shown to increase correspondance with ice thickness estimates derived using a similar snow loading approach with CryoSat-2 ice freeboard data (Petty et al., 2023c) as well as showing good agreement with a product derived directly from ICESat-2 and CryoSat-2 freeboards (Kacimi and Kwok, 2022). Due to the the lack of NESOSIM snow loading data in summer, these thickness estimates are only currently available between September and April. To increase data coverage, including the (88 °N) Arctic pole hole and to increase consistency across the observational comparisons, we use the linear interpolation/Gaussian smoothing variables from IS2SITMOGR4 as described in Petty et al (2023c) for the winter Arctic sea ice freeboard/thickness data and apply this method additionally to the Southern Ocean total ATL20 freeboard data. Differences between the raw versus interpolated/smoothed annual mean ATL20 total freeboards are provided in the Supplementary Information (Fig. S4). We analyse annual means from all months of the year, as well as the months of September and March to capture the peaks and troughs of the seasonal cycle across both hemispheres. For the Arctic thickness data, the annual mean only consists of the January-April and September to December 'winter' months, while for total freeboard we use all months of the year. These data are shown in Figs. 2 and 3.

#### Deleted: is

rormatted: Justified, Space After: 6 pt

Deleted: kg/m³

Deleted: vs

Deleted: rigures

Deleted: (a) Annual Freeboard (b) Mar Freeboard (c) Sep Fr

(d) Sep-Apr Thickness (e) Mar Thickness (f) Sep Tr

(g) Annual Area (h) Mar Area (i) Sep

Figure 2: (a, b, c) Total freeboard from ICESat-2 ATL20 v4, (d, e, f) sea ice thickness from ICESat-2 IS2SITMOGR4 v3, and (g, h, i) sea ice concentration from OSI SAF (bottom) for annual, March and September means in the 2018 to 2024 ICESat-2 period. The sea ice thickness annual mean only includes data between September and April due to data availability. The hatchings in the top two rows indicate grid-cells not included in the 'perennial ice' mask as data are missing from at least one year in the 2018 to 2024 period. Freeboard and thickness data are the interpolated/smoothed variables.

## 2.2.2 Bulk ice density estimates

The above ICESat-2 thickness retrievals follow the approach of several other studies in assuming a fixed bulk ice density, in this case 916 vkg m<sup>3</sup> – the density of pure ice. In reality, sea ice is a complex mixture of pure ice and brine, which increase bulk ice density, but also air pockets that lower bulk density, with their relative contributions varying with the evolving ice state. Various other sea ice remote sensing studies have thus utilized a lower density for multi-year ice (882 kg m<sup>2</sup>) based on the analysis of airborne Sever expedition in-situ data prior to the 1990s by Alexandrov et al. (2010). However, these bulk ice densities have been challenged in recent studies, using values inferred from multi-sensor airborne profiles (J22, Jutila et al., 2022), multi-sensor satellite methods (Shi et al., 2023) and from in-situ data collected during the Multidisciplinary drifting

Deleted: (top),

Deleted: (middle)

Deleted: A

Deleted: kg/m<sup>3</sup>

Deleted: is

Deleted: kg/m

Formatted: Font: Not Italic

Formatted: Space After: 0 pt

Observatory for the Study of Arctic Climate (MOSAiC) campaign (Salganik et al., 2024; Zhou et al., 2024). All of which generally show higher densities, linked to the younger ice state and issues with previous ice density measurement approaches.

The IS2SITMOGR4 v3 dataset also includes bulk ice density estimates calculated using the J22 empirical bulk ice density parameterization, an exponential function of the local ice freeboard

derived from coincident laser scanning, snow radar, and electromagnetic induction sounding data, (this is calculated using total freeboard minus snow depth in the ICESat-2 processing). We use these to provide an alternative, and seasonally variable, bulk ice density estimate to compare with the model results and provide added context to both the model and remote sensing-based estimates. These J22 densities are expected to be higher than the pure ice density approximation of 916 kg m<sup>3</sup>, especially for first-year ice regimes. The J22 parameterization has not been validated across different regions and seasons of the Arctic so we consider these highly experimental and use them here with caution.

Figure 3: As in Fig. 2 but for the (a, b, c) Southern Ocean total freeboard and (d, e, f) sea ice concentration

# 2.2.3 Sea ice area from satellite passive microwave

We use sea ice concentration estimates from the European Organisation for the Exploitation of Meteorological Satellites (EUMETSAT) Ocean and Sea Ice Satellite Application Facility (OSI SAF), specifically OSI-450-a, which is the third major version of the OSI SAF Global Sea Ice Concentration Climate Data Record (OSI SAF, 2022a) and OSI-430-a which is an operational extension of this product with a latency of 16 days, currently for the period 2021 onwards (OSI SAF, 2022b). We use the monthly mean concentration estimates from both datasets across the period November 2018 to April 2024 (the ICESat-2 study period). The data are posted on a 25 km x 25 km Equal-Area Scalable Earth (EASE) 2.0 grid meaning all grid-cells have a fixed area of 625 km², which we multiply by the grid-cell concentrations to derive sea ice area, before averaging across

**Deleted:** derived from coincident laser and EM-scanning airborne data

Deleted: kg/m<sup>3</sup>

Deleted:

Formatted: English (US), Do not check spelling or grammar

Deleted: Figure
Deleted: top

Deleted: and

Deleted: (bottom)

Deleted: is

basins. As in the total freeboard/thickness data, we take annual means (all months of the year), as well as September and March means across the 2018 to 2024 period.

#### 2.2.4 Observational uncertainties

An important consideration when using observations to evaluate climate models is the observational uncertainty. However, the characterization of uncertainties within sea ice remote sensing products generally focuses on grid-scale uncertainties, estimated primarily using theoretical assessments (e.g. propagation of uncertainties, Giles et al., 2007) or comparisons with ground-truth/imagery (Kern et al., 2022). These uncertainties are generally considered random/uncorrelated at the typical grid-scales they are disseminated at (~10r100 kilometres) and thus theoretically reduce to zero when averaging at basin-scales. Product or algorithm differences that drive regional-scale (~100-10.000 kilometres) systematic uncertainties are rarely accounted for. As such, the approach often used in sea ice climate model diagnostics to estimate observational uncertainty is to calculate differences in hemispheric mean sea ice area/extent across available products or algorithms (Notz & SIMIP Community, 2020; Roach et al., 2020). For sea ice concentration, multiple well established sea ice concentration/area products exist (e.g. Bootstrap: Comiso et al., 1997, NASA Team: Cavalieri et al., 1996, and OSI SAF: Lavergne et al., 2019) which enables such an approach, although this still has its limitations due to the limited (typically three product) sample size and the fact NASA Team data have a well reported low concentration bias, especially in summer (Kern et al., 2019, 2022). Recent

efforts in other domains have explored the creation of observational ensembles to better sample the full product spread (Lenssen et al., 2024), however to our knowledge no such effort has been undertaken for any of the sea ice metrics used here.

For this study, we instead estimate the basin-scale uncertainty through an evaluation of published values in SIMIP2020 and Roach et al., (2020). We use both a 'high' and 'low' uncertainty estimate to explore the impact of the observational uncertainty estimate on our model assessments considering the challenge of prescribing observational uncertainty from these limited product 'ensembles'. For passive microwave area, we use the assumption that a 0.5 million km² basin-mean sea ice area uncertainty represents a best-case 'low' uncertainty, while 1 million km² represents a less optimistic 'high' uncertainty. Uncertainty quantification is more challenging for the ICESat-2 total freeboards due to the lack of alternative freeboard products available to assess the product spread. We instead provide 'high' and 'low' uncertainty estimates based on a review of the relevant literature. The primary validation of ICESat-2 sea ice height and freeboard with coincident airborne altimetry data from NASA's Operation IceBridge show very high accuracies in the sea ice heights and total freeboard errors (10 km along-track means) of less than a few centimetres depending on the methodology used, generally indicative that ICESat-2 was likely to be satisfying the mission objectives of <3 cm freeboard uncertainty at those scales (Kwok et al., 2019). No summer Arctic or Antarctic validation analyses have been completed to-date. The summer Arctic is expected to pose more challenges due to the presence of melt ponds, but benefits from more openings in the ice cover, reducing the need to interpolate sea surface heights over large distances. In addition, an analysis of monthly mean sea surface height differences between coincident

Deleted: s

Deleted: s of

Deleted: large

Deleted: basin-scale

Formatted: Font: Not Italic

Formatted: Font: Not Italic

Deleted: has

ICESat-2 and CryoSat-2 showed overall mean differences of less than 1 cm (Bagnardi et al., 2021). For our analysis we proceed

with the assumption that a 1.5 cm basin-mean total freeboard uncertainty represents a best-case 'low' uncertainty, while 3 cm represents a less optimistic 'high' uncertainty.

**Table 3:** Hemispheric monthly mean observational uncertainty estimates applied in our study.

| Variable                             | Low uncertainty estimate | High uncertainty estimate |
|--------------------------------------|--------------------------|---------------------------|
| Sea ice area (million km²)           | 0.5                      | 1.0                       |
| Total freeboard (cm)                 | 1.5                      | 3.0                       |
| Winter Arctic sea ice thickness (cm) | 15                       | 30                        |

Winter ICESat-2 Arctic sea ice thickness uncertainty quantification is a less direct measurement than total freeboard, so the relative uncertainty increases significantly through the introduction of additional input assumptions related to snow loading and bulk ice density. Intercomparisons of ICESat-2 and CryoSat-2 winter Arctic sea ice thickness have been undertaken, showing mean differences in mean monthly winter Arctic ice thickness of ~10-30 cm between ICESat-2 and the various CryoSat-2 thickness estimates, similar to the comparisons with independent ice draft estimates obtained from upward looking sonar in the Beaufort Sea (Petty et al., 2023c). Comparisons of monthly winter Arctic mean thickness estimates between ICESat-2 and the Alfred Wegener Institute (AWI) CryoSat-2/SMOS product have also been presented in recent NOAA Arctic report cards (Meier et al., 2023, 2024), showing similar basin-scale monthly mean differences. For our analysis we proceed with the assumption that a 15 cm basin-mean winter Arctic sea ice thickness uncertainty represents a best-case 'low' uncertainty, while 30 cm represents a less optimistic 'high' uncertainty estimate. These uncertainty choices are summarized in Table 3.

# 2.2.5 Ancillary data

We use an Arctic Ocean region mask (Meier and Stewart, 2023) to analyse ICESat-2 data within an Inner Arctic Ocean domain that includes the Central Arctic, Beaufort Sea, Chukchi Sea, East Siberian Sea, Laptev Sea and Kara Sea, as in Petty et al., (2023c, Fig. 5). Focusing on the Inner Arctic avoids challenges of interpretation in the more marginal seas of the Arctic and mitigates issues with more uncertain marginal ice representation in the thickness observations, especially related to snow loading. We refer to these results as 'Arctic Ocean' throughout. We do not apply any regional masking to the Southern Ocean analysis. In addition, region masking is only applied to the ICESat-2 total freeboard and thickness data as the concentration data are considered more reliable in the marginal zones. We discuss the impact of this regional masking in the discussion.

Formatted: Caption, Line spacing: 1.5 lines

**Deleted:** Table 2: Hemispheric monthly mean observational uncertainty estimates applied in our study.

Deleted: U
Deleted: L
Deleted: S
Deleted: an

Deleted: 2

Deleted: AWI

## 2.3 Methods

## 2.3.1 Perennial ice masking

Our initial evaluations highlighted the challenge of model-observation assessments related to contrasts in total freeboard and thickness data coverage, especially as the ICESat-2 observations employ various filters, e.g. the 50% concentration and 25 km coastal filter to improve data quality, and data drop-out due to environmental factors such as clouds. Spurious model performance was also observed in the more marginal seas and in the comparisons of models with vastly underestimated ice cover in certain seasons. To enhance confidence in our ICESat-2 comparisons we thus employ a 'perennial coverage' masking to both the freeboard and thickness observations form ICESat-2, as follows:

- For the gridded freeboard/thickness observational data, calculate the fractional grid-cell coverage over time for each
  month across the 2018 to 2024 time period. Arctic data within the Inner Arctic Ocean region described above.
- Flag grid-cells as 'perennial' if they include data every year across our study period.
- Set all monthly model grid-cell freeboards and thickness to zero (instead of NaN) across valid regridded ocean/sea ice grid-cells.
- For every month, mask all model grid-cells (set to NaN) outside the perennial mask for the relevant metric.
- Calculate monthly, hemisphere mean quantities grid-cells for the observations and the models.
- Calculate annual means by taking the annual average of the monthly 'perennial' means.
- For the spatial comparisons, only compare data across the 'perennial' grid-cells.

The impact of this perennial masking is that our annual freeboard/thickness results are likely to be skewed high as the observations do not include regions of low (<50%) concentration ice. Clouds/data gaps are generally more likely in the more marginal seas also. The freeboard/thickness analysis is thus more of an assessment of ice conditions where we have consistent ICESat-2 data coverage. The assessment of simulated sea ice coverage is addressed more comprehensively with the passive microwave sea ice concentration/area data that provide full coverage data across both hemispheres.

# 505 2.3.2 Internal variability and plausibility estimates

To characterize internal variability we calculate, for each model, the standard deviation of the given metric of interest (e.g. total freeboard) averaged over the given region and the time window of interest (e.g. annual mean or a given month for the 2018 to 2024 ICESat-2 period) across all available ensemble members. As in SIMIP2020 we apply the Bessel correction to estimate an unbiased population standard deviation from a sample, accounting for the variable ensemble size across CMIP6. To explore the sensitivity to ensemble size and our chosen time-period, especially for models with low (or no) ensemble size, we also calculate 6-year running means within a larger time window (start years of 2015 through to 2024, ensuring we analyse only the SSP2.4.5 runs where the ensembles are consistent). We choose this window size as a balance between the benefits of

A climate model is not expected to exactly match the time period of our own reality due to the impact of internal variability.

Deleted: 1

Deleted: IS-2

Deleted: that

Deleted: s

Deleted:

Deleted:

increased sampling and the cost of increasing the likelihood of trend contamination. We additionally calculate a CMIP6 mean internal variability for each given metric by taking the mean internal variability across all models for those with at least 5 members. We follow SIMIP2020 and define a plausible range as:

$$P = +/-2\sqrt{(\sigma_{int}^2 + \sigma_{obs}^2)} \tag{6}$$

Where  $\sigma_{int}$  is the internal variability and  $\sigma_{obs}$  is the observational uncertainty. The factor 2 effectively provides a 95% plausibility window based on the observational uncertainty and internal variability. In addition, following earlier CMIP analyses (Olonscheck and Notz, 2017; Santer et al., 2008), we calculate a model plausibility index as:

$$\phi = (\mu_{mod} + \mu_{obs}) / \sqrt{(\sigma_{int}^2 + \sigma_{obs}^2)}$$
 (7)

where  $\mu_{mod}$  and  $\mu_{obs}$  are the mean model and observational quantities of interest. A plausibility index of zero indicates perfect agreement between the model and observation, with higher values in either direction, based on the direction of the model bias, indicating worse agreement and lower plausibility. We use this plausibility index to compare plausibility across metrics.

#### 2.3.3 Spatial assessments

For our freeboard and thickness spatial comparisons, we calculate a Mean Absolute Error (MAE) by comparing the perennial grid-cell means between the observations and models. For area, we choose Sum Absolute Error (SAE) to express the results in units more consistent with the total area metric. We then additionally estimate spatial internal variability, following the same approach as above, but at the grid-cell level following the approach of Schaller et al., (2011) for regional precipitation assessments. The regional estimate of internal variability is used to provide context to our regional bias assessments, However, the ability of models to accurately represent the regional manifestation of sea ice internal variability is unclear at best, so we provide this analysis with caution and do not use the spatial comparisons to assess model plausibility explicitly. We use only the perennial masked data in this analysis to increase confidence in our comparisons. Due to the short time period and increasing/uncertain role of internal variability in the spatial assessments, we do not assess pattern correlation, as in other regional assessments (Watts et al., 2021). We instead provide a composite regional bias analysis to provide further insights into typical spatial sea ice bias patterns simulated by CMIP6 models.

#### 3 Results

# 3.1 Bulk ice density and total freeboard analysis

We first assess bulk ice density and total freeboard across the 1½-model subset that provide direct outputs of ice freeboard. As discussed in Sect. 2.1, a key variable in the conversion between ice thickness and freeboard is the bulk ice density which we estimate using the two methods described in Sect 2.1.: Method 1½ using the provided variables of freeboard, snow depth and ice thickness and a prescribed snow density of 330 kg m<sup>3</sup>. Method 2½ using the sea ice mass and volume outputs. Our initial

Deleted: increasingly

Deleted: to generate a

**Moved down [2]:** We use only the perennial masked data in this analysis to increase confidence in our comparisons.

Deleted:

**Deleted:**, following the approach of (Schaller et al., 2011) for regional precipitation assessments

Deleted: T

Moved (insertion) [2]

Deleted: 6

Deleted: Section

Deleted:

Deleted: kg/m<sup>3</sup>

Deleted:

analysis produced notably high bulk ice densities close to that of pure seawater in the ACCESS-CM2 model (see Supplementary Information, Fig S5), especially in the Arctic (Method 1/Method 2: 1005/940 kg m<sup>-3</sup> for the Arctic and 955/975 kg m<sup>-3</sup> for the Southern Ocean, both with significant seasonal variability). We were unable to ascertain the cause of these anomalously high densities and thus dropped this model from the rest of our freeboard subset analysis. We suspect an underlying error in how freeboard was calculated in post processing, as other key metrics did not display such anomalous behaviour.

Figure 4: Derived mean (a) Arctic Ocean and (b) Southern Ocean bulk sea ice density in the 16 model CMIP6 subset (models with available freeboard, snow, and ice thickness data, not including ACCESS-CM2). Circles indicate the annual mean for each model and horizontal lines show the standard deviation across months (a proxy for the seasonality). A 50% SIC masking is applied before spatial averaging. The dashed vertical black line shows the 916 kg m³ 'pure ice' bulk ice density assumption used in the IS2SITMOGR4 dataset. Also shown is the J22 ice density from IS2SITMOGR4 averaged between September and April 2018 to 2024.

In Fig. 4 we show estimates of bulk ice density (mean and seasonal variability) for the reduced 16-model subset across both the Arctic Ocean and Southern Ocean over our ICESat-2-study period of 2018-2024. The Method 2 (simass/sivol, Eq. 4) results all show no seasonality and fixed values of 900 kg m<sup>-3</sup> (MRI-ESM2-0), 910 kg m<sup>-3</sup> (MPI and CNRM models), 916 kg m<sup>-3</sup> (the remaining models). Note that CNRM-CM6-1-HR did not provide the needed output (Table 2), but we expect that to feature the same density as the other CNRM models. For Method 1 (ice freeboard/hydrostatic, Eq. 5) several models show no seasonal variability as ice density is simply fixed (no variable internal temperature or salinity) in their model setup, and these values were generally consistent with the Method 2 results. However, several Method 1 model estimates produced significant ice density seasonality, which we expect is due to inclusion of a variable internal ice temperature and salinity physics scheme in the more advanced sea ice models. Interestingly, the bulk ice densities in the seasonally variable Method 1 models were notably higher or lower than the Method 2 results and the pure ice density value. More specifically, the CNRM models produced densities of 898 ± 8 kg m<sup>-3</sup> (Arctic) and 883 ± 8 kg m<sup>-3</sup> (Southern Ocean) while the MRI-ESM2.0 model produced densities of

Deleted: kg/m³

Deleted: kg/m³

Deleted: 5

Deleted: 5

Deleted: IS-2
Deleted: 5

Deleted: 1

Deleted: 6

Deleted: +/-

Deleted: +/-

~891 ± 2 kg m³ (both hemispheres, minimal seasonal variability), all consistently lower than the Method 2 results and the pure ice/average bulk ice densities across both hemispheres. In contrast the CESM2 models produced densities of ~925 ± 3 kg m³ (both hemispheres) and the NorESM2 models produced densities of ~925 ± 3 kg m³ (Arctic) and ~922 ± 3 kg m³ (Southern Ocean) consistently higher than the respective Method 2 results and the pure ice/average bulk ice densities. Centres with multiple model configurations (CNRM, NorESM2 and CESM2) produced bulk ice densities largely consistent across their respective configurations, as expected. At least one modelling group (CESM2) confirmed there are effectively two bulk ice densities in the model, with the Method 1 density reflecting the internal ice physics and the density used in the freeboard calculation, but the Method 2 densities based on fixed salinity/temperature, used in atmospheric coupling assumptions (D. Bailey, personal communication). We expect that similar discrepancies may be the cause of the differences observed across the other models.

Figure 5: Comparison of the mean (a) Arctic Ocean (b) and Southern Ocean total freeboard in the 17 model CMIP6 freeboard subset from the ensemble mean total freeboard output (circles), derived total freeboard assuming a fixed density of 916 kg m³ (triangles), and total freeboard observations from ATL20 v4 (processed as discussed in Sect. 2 and 3) (red) for the LCESat-2 study period (2018-11 to 2024-04). Horizontal lines show the seasonal variability (monthly standard deviation).

Also shown in Fig. 4 is the bulk ice density inferred from the J22 empirical density parameterization applied to ICESat-2 data for the Arctic only (September to April data only due to thickness data availability), showing mean Arctic bulk ice densities of ~931  $\neq$  3 kg m<sup>3</sup>. These densities are notably higher than the other estimates, although within the seasonal range of the CESM2/NorESM2 model Method 1 results and similar to values for bulk density quoted in recent studies (see discussion in Sect. 2.2.2). A more detailed assessment/validation of the bulk ice density estimates is considered beyond the scope of this

Deleted: +/Deleted: +/Deleted: +/Deleted: +/-

Deleted:

Deleted: mean

Deleted: left

Deleted: (right)

Deleted: 6

Deleted: K
Deleted: IS-2

Deleted: +/-

study considering the significant scale differences between GCMs and field studies. Despite the lack of consensus in methodology, the multi-model annual mean from both methods across the 16-model subset is 912-914 kg m<sup>-3</sup>, close to the pure ice bulk ice density assumption (dashed line, discussed in Sect. 2.2).

Next, we briefly explore the impact of differences in bulk ice density on estimates of total freeboard. In Fig. 5 we show estimates of total freeboard from the 17-model subset from both the direct total freeboard output (grid-cell mean snow thickness added to the ice freeboard output) and with total freeboard calculated as in Eqs., 2 and 3 using our own prescribed estimate of bulk ice density (916 kg m<sup>-3</sup>). We also show observational estimates of total freeboard from ATL20 (using the interpolated/smoothed and perennial analysis) for reference, a more detailed plausible range assessment is provided in the following sections. The results in Fig. 5 show multi-model annual mean total freeboards slightly lower than ATL20 (22 cm vs 27 cm annual means respectively), with negligible differences using the prescribed/fixed ice density method compared to the direct output method (differences of < 1-2 cm). Note that the ACCESS-CM2 model shows a significant increase in freeboard when using the prescribed ice density method, bringing those freeboards in much better agreement with ATL20, further suggesting issues with the underlying density/freeboard calculation. The MRI-ESM2.0 model shows the biggest difference in total freeboard from the two methods in the 17-model subset (~3 cm difference), with our prescribed density reducing the total freeboard estimate as this was the model with the lowest estimated bulk ice density (~890-895 kg m<sup>-3</sup>). The strength of the seasonal cycle (the interannual variability contribution to the total monthly variability across this period is low, not shown) appears broadly consistent between the models and ATL20 for the Arctic Ocean results (standard deviation of ~6 cm for the multi-model mean and ATL20), however in the Southern Ocean the ATL20 monthly total freeboard variability is noticeably lower (~3 cm) compared to the multi-model mean (~6 cm for the multi-model mean). The prescribed versus variable bulk ice density freeboard estimates do not produce a significant difference in the strength of the freeboard seasonal cycle. A more detailed plausibility assessment accounting for observational uncertainty and internal variability is provided in the following sections. Overall, this bulk ice density and total freeboard analysis suggests the use of a fixed bulk ice density to calculate and analyse total freeboard across the wider 36-model subset that provide the necessary outputs of sea ice thickness and snow depth (Table 2) introduces only small additional uncertainty (<1-2 cm) and ensures consistency in the density approximation used between models and observational freeboard estimates. We discuss the implications of this approach and suggestions for the wider community in the later discussion.

#### 3.2 Plausibility assessments

Next, we explore the plausibility of the full CMIP6 suite, utilizing total freeboard metrics in addition to sea ice area across both hemispheres and winter Arctic sea ice thickness. We undertake this analysis for annual means and the two months that broadly represent the seasonal sea ice maximum and minimum months (September and March), with sea ice thickness data only included between September and April in the annual means due to data availability, as discussed in Sect. 2.2. To assess the plausibility of model output, we need to consider both the internal variability and observational uncertainty of our chosen

Deleted: 5
Deleted: Section

Deleted: reduced 15

Deleted: ¶
Deleted: Annual
Deleted: 5
Deleted: 8

Deleted: in our initial analysis,

Deleted: (not shown) showed

Deleted: 15

Deleted: 1

Deleted: A
Deleted: Section

metrics, which were discussed in detail in <u>Sect. 2</u>. Our estimates of Arctic Ocean mean total freeboard internal variability for March, September and <u>annual</u> time periods for the 13 models with at least 5 ensemble members are shown in Fig. 6.

Figure 6: Estimates of internal variability (1 ensemble standard deviation, with Bessel correction) of Arctic Ocean total freeboard from ATL20 v4 (processed as discussed in Sect. 2 and 3) for the (a) annual, (b) March and (c) September periods for the 6-year mean over the JCESat-2 study period 2018-2024 (black lines) and running means across the longer 2015-2029 period (red crosses) for all the CMIP6 models with at least 5 ensemble members. Multi-model CMIP6 mean values shown on the right.

Our results show a significant spread in internal variability estimates across the different models, despite the application of the

Bessel correction to account for sample size, with values ranging from ~ 1 cm to 6 cm depending on the CMIP6 model and time period of interest analysed. Similar spread was noted in previous studies (Notz & SIMIP Community, 2020; Roach et al., 2020). While the inter-model internal variability spread is high, the differences are largely consistent across the three seasonal time periods. Using the wider time window to increase sampling (2015 to 2029, 10 rolling 6-year time means) provided a moderate impact on the internal variability estimates with ~1 cm on average higher values for the annual, March and September

Deleted: Section

Deleted: Annual

Deleted: IS

Deleted: and

Deleted: a

CMIP6 mean internal variability estimates. At the individual model level, impact was highly variable, with some models showing no significant change, most models showing a moderate increase, and UKESM1-0-LL especially showing a significant increase. Our multi-model mean estimates of Arctic Ocean total freeboard internal variability using the 2015-2029 window are ~2.5 cm (annual), 2.2 cm (March), 3.6 cm (September). Similar multi-model internal variability differences were found across our other metrics and hemispheres, with the multi-model means for all metrics and time-periods summarized in Table 4. For the plausible range analysis below, we proceed with using the multi-model mean internal variability estimates in Table 4. applied to all models. We include a discussion of this assumption in our summary section below.

Table 4: CMIP6 mean internal variability estimates (1 standard deviation) calculated across ensemble members and 10 rolling 6-yr means over the period 2015-2029 for the different metrics of interest across the CMIP6 multi-model subset with at least 5 ensemble members. Note that we do not analyse Southern Ocean sea ice thickness in this study and the Arctic annual mean thickness estimate does not include data between May and August (based on IS2SITMOGR4 data availability). Total freeboard CMIP6 mean values are shown in Fig. 6.

| Metric                 | Hemisphere                    | Annual      | March                        | September   |
|------------------------|-------------------------------|-------------|------------------------------|-------------|
| Sea ice area (106 km²) | Arctic Ocean / Southern Ocean | 0.38 / 0.31 | 0.35 / 0.17                  | 0.54 / 0.45 |
| Total freeboard (cm)   | Arctic Ocean / Southern Ocean | 2.4/1.0     | 2.3 / 3.3                    | 3.6 / 1.1,  |
| Sea ice thickness (m)  | Arctic Ocean                  | 0.16 / X    | <u>0.15 / X</u> <sub>▼</sub> | 0.29 / X    |

In Fig. 7 we show comparisons of annual mean Arctic Ocean total freeboard estimated from the 36-model CMIP6 subset (see Table 2), calculated using our prescribed bulk ice density (916 kg m<sup>-3</sup>) with observational estimates of total freeboard from ATL20 for our ICESat-2 study period (2018 to 2024). The analysis shows the ensemble mean for each model with a plausibility window for each model calculated following Sect. 2.3.1 based on both the high and low observational uncertainty estimates (Table 3) and multi-model mean estimates of internal variability (Table 4). The models are ranked in order of the ensemble mean differences between the model and observation for the given metric. Overall, the multi-model CMIP6 mean Arctic Ocean total freeboard (26 cm) is similar to ATL20 (27 cm), but with a large multi-model ensemble spread (~7 cm). In general, there are more models that are considered plausible (24), compared to implausible (12) at the 2 sigma, 95% confidence level. The use of a high uncertainty estimate (3 cm instead of 1.5 cm) impacts the potential plausibility of 4 of the mid-lower ranked models.

Also shown in Fig. 7 is the mean absolute error (MAE) in grid-cell annual mean total freeboard for each model vs ATL20, which captures the regional differences in the time-mean total freeboard between the models and observations. Regional-scale comparisons of climate model outputs are challenging and require deeper consideration of internal variability and model expectations, so we use ensemble mean spatial comparisons for added context of potential offsetting biases rather than as a plausibility constraint. The results show a multi-model MAE of ~8 cm compared to the < 1 cm spatial-mean multi-model mean difference, highlighting the significant role of off-setting regional differences. The non-monotonic increase in MAE down the hemisphere-mean difference ranked models alludes to the challenges of hemisphere-scale mean comparisons, although the</p>

Deleted: 8
Deleted: 5
Deleted: 3
Deleted: 3

Deleted: M
Deleted: 0.39 / 0.30

Formatted: Caption

**Deleted:** 0.36 / 0.18 **Deleted:** 0.53 / 0.44

Formatted: Superscript

Deleted: 2.4 / 1.0

**Deleted:** 2.2 / 3.3 **Deleted:** 3.5 / 1.1

Deleted: 0.16

Deleted: 0.15

Deleted: 0.28

Deleted: Table 3: CMIP6 mean internal variability estimates (1 standard deviation) calculated across ensemble members and 10 rolling 6-yr means over the period 2015-2029 for the different metrics of interest across the CMIP6 multi-model subset with at least 5 ensemble members. Note that we do not analyse Southern Ocean sea ice thickness in this study and the Arctic Annual thickness estimate is composed only of September to April data (based on IS2SITMOGR4 availability). Total freeboard CMIP6 mean values are shown in Figure 6.

Deleted: 1

Deleted: IS

Deleted: 2

Deleted: 3

lowest ranked models still generally show higher MAE compared to the higher ranked models alluding to consistent 750 hemisphere-scale biases in those models. We provide a more detailed plausibility assessment of regional biases in Sect. 3.3.

Figure 7: (a) Comparisons of annual mean (2018 to 2024) Arctic Ocean total freeboard estimates from the 37-model CMIP6 subset (ensemble means, using a fixed ice density) and observations from ICESat-2 ATL 20 v4 (processed as discussed in Sect. 2 and 3). Horizontal lines on each model show the plausibility window based on internal variability and both the low (bars) and high (whiskers) observational uncertainty estimates. The red circle and vertical red line shows the ATL20 observational mean. ES: ensemble spread. b Mean Absolute Error (MAE) of the spatial differences across the 25 km x 25 km grid-cells.

Formatted: Space After: 0 pt

Figure 8: As in Fig. 7, but for the Southern Ocean.

In Fig. 8 we show the same annual mean total freeboard plausibility analysis but for the Southern Ocean. In this case, the 37-model mean Southern Ocean total freeboard (16 cm) is significantly lower than ATL20 observations (26 cm) and exhibits more significant multi-model ensemble spread (~9 cm). In general, there are more model estimates of Southern Ocean total freeboard that are considered implausible (29), compared to plausible (7) at the 2 sigma, 95% confidence level. The use of the high observational uncertainty estimates results in only three more models being considered plausible at this confidence level. This result was widely expected considering the strong sea ice biases reported in earlier Southern Ocean CMIP6 analyses (Roach et al., 2020). The Southern Ocean total freeboard MAEs are also higher than the Arctic (~14 cm multi-model mean). The strong overall negative bias across most of the CMIP6 models still appears to dominate the MAE contribution but is again explored more in Sect. 3.3.

Figures showing the same plausibility analysis for both hemispheres across all other metrics and selected months (September and March) are provided in the Supplementary Information (Figs. S6 to S18). We instead summarize results from all 15 combinations of metrics, time periods, and hemispheres into a single analysis to more efficiently assess CMIP6 model performance. For this analysis we use the plausibility index described in Sect. 2.3.1. that represents the plausibility of a given model's ensemble mean for the given metric considering internal variability, observational uncertainty and the direction of the overall model observation bias. For the Southern Ocean analysis, we use the 'high' observational uncertainty estimate due to added observational complexities discussed earlier and noted in our comparisons and to ensure more models can be included in a constrained subset analysis.

Deleted: Figures

Formatted: Space After: 0 pt

This plausibility index analysis is shown in Fig. 9, with models ranked by the mean plausibility index averaged across all 15 different assessment combinations (columns in Fig. 9). We also highlight the models and metrics where the plausibility index is > \$\pm\$ 3, which is equivalent to a 99% confidence level that we can consider those models implausible based on our chosen criteria. Four of the models do not provide freeboard/thickness output and are thus included at the bottom of the figure. The CMIP6 model ensemble mean results show some of the best plausibility scores, including plausibility indices 

Figure 9: Plausibility assessment of the full 40 model CMIP6 suite for the individual model ensemble means and the multi-model CMIP6 mean, (top row) across all metrics, time-periods and hemispheres. The plausibility index (φ) is calculated as in Eq. 7 using the 2015-2029 internal variability estimates and observational uncertainties listed in Table 3 (low uncertainty for the Arctic, high for the Southern Ocean). Models are sorted by the average φ across all values shown, with the four models missing freeboard/thickness output included at the bottom and the CMIP6 mean included at the top. Lower values are considered more plausible. Values of φ greater than 3 are outlined in black.

Several models show plausible results for all metrics except one, most frequently the March Southern Ocean total freeboard, and all four of these models show similar negative plausibility indices (-3.1 to -3.5) indicating a slight negative model freeboard bias in austral summer. It is also interesting to note that the March Southern Ocean area results (around the Antarctic sea ice

Deleted: 2

minimum) are typically considered more plausible than the September and annual mean results, but this pattern generally reverses for freeboard. This result is driven partly by the low sea ice coverage in Southern Ocean sea ice in March limiting the magnitude of the bias and also our estimates of internal variability, especially for area – the lower the internal variability the lower the plausibility threshold. This analysis raises important questions about the ability of current climate models to reasonably constrain more seasonal internal variability estimates, especially for low sea ice months when non-linear/non-Gaussian variability is likely. We discuss these concerns more in Sect. 4.

Overall, this combined plausibility analysis provides what we consider a useful framework for evaluating sea ice model output using a series of metrics and relevant observations. Caution should be used in applying such plausibility results, however, as results are sensitive to choices regarding internal variability and observational uncertainty, as well as our chosen metrics. Model plausibility and constraint assessments depend ultimately on the overall goal. In the following section we provide an example of CMIP6 model constraints based on this plausibility analysis, focusing on constraining modern-era CMIP6 sea ice simulations.

#### 3.2.1 Impacts of model plausibility constraints

In Fig. 10 we show the impact of plausibility constraints on CMIP6 estimates of the seasonal cycle in sea ice volume across both the Arctic and Southern Ocean. The same plot but for the individual metrics of sea ice area, total freeboard and thickness is shown in the Supplementary Information (Fig. S19). The constrained model subset is generated independently for each hemisphere, with the assumption being that a model that performs well in one hemisphere should not be used to judge performance in the other hemisphere due to differences in priorities and model development efforts. We produce the constrained subsets in this example using just the annual mean plausibility indices and require a model to have a plausibility index < 3 for all available variables (area, total freeboard, and thickness for the Arctic, and area and freeboard for the Southern Ocean). We repeat this analysis for just the area metric, then for area, freeboard and thickness (thickness for the Arctic only) to assess the relative benefits of ICESat-2 data in this example constraint analysis. The goal is to explore how our plausibility constraints impact the CMIP6 multi-model mean and spread and focus here on the longer 2015 to 2035 time period. For the Arctic Ocean this results in an area constrained subset of 31 models and an area, freeboard and thickness constrained subset of 27 models. For the Southern Ocean this results in an area constrained subset of 21 models and an area and freeboard constrained subset of 14 models. We do not include the four models that only provide area (see Table 2) in either subset.

For the Arctic, the area constrained volume seasonal cycle shows a similar inter-model spread (shading in Fig. 10) and a slight increase in sea ice volume ( $\sim 1 \times 10^3_{\rm s}$  km<sup>3</sup>) across all months. When constraining also with freeboard and thickness, the intermodel spread reduces considerably (by  $\sim 50$  %) and the volume across all months becomes slightly lower ( $\sim 0.1 \times 10^3$  km<sup>3</sup>) than even the unconstrained volumes, especially in summer. For the Southern Ocean, the area constrained volume seasonal cycle shows a reduced inter-model spread (again by  $\sim 50$  %) and a more significant increase in sea ice volume across all months ( $\sim 1.4 \times 10^3$  km<sup>3</sup>), especially in austral winter. When constraining also with freeboard, the inter-model spread shows a small extra

Deleted: Annual

Deleted: 1

Deleted: M

Formatted: Superscript

Deleted: M

Deleted: M

reduction in some months and the volume shows a slight additional increase compared to the area constrained results (0-1x 10<sup>3</sup> km³). Constraining models also with the March and September Arctic Ocean plausibility results (

Figure 10: Seasonal CMIP6 ensemble mean sea ice volume (lines and circles) and ensemble spread (shading, one standard deviation) from the unconstrained 36-model subset (black), the annual area constrained subset (blue), and the subset constrained with all considered annual variables, including area and freeboard for both hemispheres and sea ice thickness for the Arctic only (red).

Overall, this basic analysis highlights the significant additional impact of constraining CMIP6 output with ICESat-2 freeboard and thickness estimates compared to sea ice area.

## 3.3 Regional plausibility assessments

We primarily explored hemisphere-mean comparisons in the preceding sections, but larger differences can be observed when analysing model differences at more regional scales. Comparing models and observations at the grid-cell level can be Deleted:

Deleted: M

Deleted: two

Deleted:

Deleted: the addition of

misleading as fully coupled climate model runs are not intended to perfectly capture our current reality but instead simulate expected climatic conditions within ideally a reasonable range of expected internal variability. To explore the ideas of regional plausibility we follow the approach of other studies (e.g. Schaller et al., 2011) and attempt to characterize internal variability at the grid-scale, towards a regional plausibility assessment as described in Sect. 2.3.3. As we assume the observational uncertainties in the hemisphere-mean analysis were driven primarily by biases or systematic error in the observations, we apply these same error estimates at each grid-cell as a first-order uncertainty approximation. In reality, we expect a more complex regional combination of different error contributions.

Figure 11 shows maps of the annual Arctic Ocean total freeboard difference between the 36-model CMIP6 subset and ATL20. The hatchings indicate regions that are considered implausible based on the regional plausibility index. The regional internal variability estimate calculated from the CMIP6-mean ensemble spread from models with at least 5 ensemble members is shown also (2 standard deviations), showing a mean value of ~4-8 cm across the Arctic, with some small increases along the coast/ice edge. Note again that the 3 cm total freeboard uncertainty is added to this internal variability to produce the regional plausibility threshold. The models are ranked by the overall MAE and help visualize the regional contributions to the overall bias and regions of implausibility. In the better performing models, specific regions appear with some consistency across the models, including a few models with negative freeboard anomalies in the thicker Central Arctic regime along the Greenland/Canadian Arctic coast, as well as the Chukchi Sea, and some models with positive anomalies North of Greenland and the Beaufort Sea.

Many of these regions represent key dynamical features of the Arctic sea ice cover, e.g. the Beaufort Gyre (Petty et al., 2016)

For the lower performing models, the regions of implausible bias extend further across the Arctic and allude to broader thermodynamic drivers of the biases. A full thermodynamic/dynamic accounting for the regional differences is considered beyond the scope of this study but could form the basis of future model development evaluation methods. Difference maps of the remaining Arctic Ocean metrics are shown in the Supplementary Information (Figs. S20 – S31). In general, the sea ice area biases are more prevalent in the marginal seas, especially in March as expected, while the regional distribution of biases and implausibility in thickness align closely with these total freeboard results, further highlighting the potential for total freeboard to provide useful regional diagnostics as a proxy for thickness biases. Similar regional bias patterns in CMIP6 models were observed in Watts et al., (2021), which included comparisons to CryoSat-2 and the earlier ICESat mission.

and the Wandel Sea (Schweiger et al., 2021) and allude to deficiencies in atmospheric dynamics (e.g. wind forcing).

Deleted: was

Deleted:

Deleted: with

Deleted: Figures

Figure 11: Total freeboard difference of the 2018 to 2024 annual mean for each CMIP6 model (ensemble mean) relative to ATL20 v4 (processed as discussed in Sect. 2 and 3). Hatchings indicate regions that are considered implausible based on the regional plausibility index analysis. Models are shown in order of lowest (top left by row) to highest mean absolute error (MAE). The dashed line (50°E and 130°W meridians) is used for the Eastern and Western sector analysis. Note that only model grid cells within our Inner Arctic Ocean domain and which are 'perennially ice-covered' in ATL20 v4 are included here. Final panel shows our estimate of CMIP6 mean regional internal variability (two standard deviations, Sect. 2.3.3.)

We next use regional composites, grouping the models based on the biases measured within the Eastern and/or Western Arctic (denoted by the dashed line in Fig. 11 from 60 °E to 120 °W), to highlight the predominant regional manifestations in freeboard and thickness biases seen in our spatial difference maps. Figure 12 shows composite maps and difference plots based on

Formatted: Font: 9 pt

Deleted: Next

Deleted: we

Deleted: Figure

Deleted:

different combinations of the direction of the total Eastern Arctic (East) vs Western Arctic (West) bias for both freeboard and thickness (annual means). Note that for this annual mean Arctic Ocean total freeboard comparison, there were no models that showed a -East and +West bias.

Figure 12 (top row) Annual multi-model CMIP6 ensemble mean total freeboard for all models (a) then composites based on their East/West bias pattern (b to d). (second row) top row minus the ATL20 v4 annual mean total freeboards (processed as in Sect. 2 and 3). Third and fourth row show the same but for Sep to Apr Arctic sea ice thickness from CMIP6 and differences with IS2SITMOGR4 v3 thickness data. Stippling indicates where at least 80% of the models in the respective composite agree with the sign of the difference.

For freeboard, omodels feature a +East and -West bias, 1 models feature a +East and +West bias and 17 models feature a -East and -West bias. The +East and -West bias models show generally consistent freeboard across the Arctic and thus fail to capture the strong gradient from the western Arctic to the East, while the other model composites show a clear West to East

Deleted: Annual

Formatted: Font: 10 pt

Deleted: 10

Deleted: 10

Deleted: 16

freeboard gradient but with freeboards that are either too high or too low on average. For the -East and -West models, the models show better agreement with observations in the Beaufort Sea region on average but with no clear agreement across the models, again suggestive of unique dynamical challenges for models in this region. The sea ice thickness (September through April) bias composites show virtually identical spatial composite difference patterns and model agreement to the freeboard results, with only small differences in the distribution of models across the three composites. Composite maps for both September and March are provided in the Supplementary Information (Figs. S32 and S33), which show similar regional patterns and model composites, with reduced coverage and more models showing a -East and -West bias in September.

| Figure 13: As in Fig | . 11 but | for the | Southern | Ocean |
|----------------------|----------|---------|----------|-------|
|----------------------|----------|---------|----------|-------|

In Fig. 13 we show the annual mean total freeboard difference analysis for the Southern Ocean. The mean internal variability is similar to the Arctic (~ 4-8 cm), with the highest values in the western Weddell Sea, where the ice is generally thickest in the models. As shown in our basin-scale analyses, the Southern Ocean results exhibit larger overall biases compared to the Arctic. However, Southern Ocean sea ice is also generally considered more regionally variable with significant regional differences in climate driven by its unique geography (where the sea ice pack surrounds the continent of Antarctica and covers three different ocean basins) providing further motivation for such regional assessments. As in the Arctic analysis, there are strong differences across the models in where the freeboard biases manifest, but some evidence of problem areas, including the Weddell Sea and other shelf sea regions. It is also interesting to note the significant number of models that show an overall negative bias to the observations everywhere except the Ross Sea, a region of thin ice and strong wind-driven polynya activity.

We also observe more consistent negative model biases along the ice edge in the Southern Ocean analysis. These 'biases' should be treated with more caution, as wave contamination along the ice edge is a known issue with ATL20 (data are masked below 50% concentration to try and mitigate this) and is expected to impact the Southern Ocean more than the Arctic due to stronger wave activity (Horvat et al., 2020). In addition, significant implausible negative biases are observed around the Antarctic coastline in several models that should be a source of future model development focus. In additional analysis (not shown) we confirmed this was present across the native model grids and not introduced in our regridding step. In Fig. 14 we repeat the composite analysis, but as there was no clear longitudinal dependence on the biases, we simply grouped the models based on the overall direction of the mean bias. For the 7 models with positive bias, there is general agreement on the regional pattern of the bias within the ice pack, however the negative differences around the coast and the ice edge appear more model specific. The 30 models with negative bias are consistently negative except for the Ross Sea and the eastern Weddell Sea. These again suggest issues with the underlying dynamics and the response to large-scale circulation, or the simulated internal variability not fully capturing the regional patterns that would alter the plausibility of the observed biases. September and March composite maps are also shown in the Supplementary Information (Figs. S34 and S35) with the September results showing similar regional distributions, while the limited ice cover in March significantly reduces the value of the regional analysis.

Deleted: Figure

Deleted: Annual

Deleted: is

Deleted: 29

Figure 14: As in Fig. 11, but for the Southern Ocean freeboard. No regional bias composite is shown—only composites where basin—averaged differences from ATL20 v4 are either positive or negative.

# 4 Discussion

Sea ice bulk density has long been a source of uncertainty in remote sensing efforts and has arguably not been an explicit focus of global sea ice modelling efforts to-date. However, assessments of freeboard and the introduction of variable density schemes provide motivation to increase documentation and focus on the underling sea ice density assumptions or parameterization schemes used. More direct outputs of freeboard and assessments of these outputs instead of our own derived freeboards could additionally help mitigate the uncertainty introduced by our own density choices. Recent observational analyses have also alluded to significant increases in bulk ice density compared to historical estimates, linked especially to the thinner and younger Arctic sea ice pack, but also potential issues with prior data collection campaigns and interpretation (Jutila et al., 2022; Salganik et al., 2024; Shi et al., 2023; Zhou et al., 2024). This trend could continue as the Arctic (and perhaps Southern Ocean) ice pack continues to thin and lose older ice, so models with variable and realistic density parameterizations (based on prognostic internal temperature and salinity) calibrated to current day observations are encouraged.

5 Uncertainty quantification in remotely sensed sea ice products continues to be a challenge. We provide one basic approach (high vs low fixed uncertainty estimates based on a literature review) and hope that more community engagement and consensus can be undertaken in the near future. Uncertainty quantification needs to include a full accounting of possible error

Deleted: Figure

**Deleted:** There are several important caveats and issues raised by this study:

sources, including sampling/representation errors. The ongoing ESA-funded Sea Ice-thickness product iNter-comparison eXerciSe (SIN'XS) initiative (https://sinxs.noveltis.fr) aims to increase community focus on this issue. In addition, ensemble-based methods would provide a more robust framework for assessing observational uncertainty and provide important insights into the regional uncertainty estimates. Increased development and uncertainty estimates of the freeboard and thickness products in more marginal ice regimes is urgently needed, especially in the Southern Ocean where some of the model biases were more questionable. Sea ice reanalysis systems or fusion with other sensors are urgently needed.

The current time period of ICESat-2 freeboard and thickness data (2018 to 2024 at the time of writing) is short and not representative of typical climate timescales used in model assessment efforts. The short period increases the contribution of internal variability and the challenge of quantifying internal variability across our chosen metrics, especially from models with lower ensemble member counts and significant model biases. Our utilized method demonstrates a potential approach in the absence of longer timeseries (an increased window around our current time-period), with the benefit of limiting trend contamination. More years of observational data can both increase confidence in our internal variability estimates and reduce its contribution to overall uncertainty, with important implications for our plausibility assessments. We hope future work towards integration of sea ice altimetry data from NASA's ICESat (2003 to 2008) and ESA's CryoSat-2 (2010 to present) mission, together with advances in sea ice reanalyses, will provide important benefits here. Longer records will also enable assessments of thickness and volume trends, especially for the Arctic, where we have more confidence in the snow loading inputs. Regardless of the time-period, different models can produce very different internal variability estimates, which provides a further challenge and source of uncertainty we need to consider more in future work. Utilizing independent large ensembles for constraining internal variability independent of the CMIP6 suite is an alternative approach worth exploring. The ability of models to accurately capture regional internal variability is questionable, which prevents us from more confidently prescribing implausible regional biases in the models.

The summer assessments were more challenging to decipher due to the bigger role of coverage issues and differences between the models and observations. We employed a 'perennial' ice masking approach to mitigate coverage issues and improve the robustness of those comparisons, but comparing biases of global climate models in these small regions poses additional questions, e.g. how much should we expect climate models to simulate these more fractional ice packs. Other studies have mitigated this to some degree by focussing more on the strength of the seasonal cycle (Massonnet et al., 2012) which could be worth considering. Similarly, understanding seasonal snow evolution and biases and how that relates to our seasonal biases in freeboard and thickness would provide a logical next phase of this study.

Our regional analysis provided new insights into how total freeboard can be used to diagnose regional sea ice biases in models, with our East-West composite analysis providing a simple framework for assessing and grouping models. Increased focus on regional sea ice internal variability estimates in large ensembles should be explored to enhance confidence in the regional plausibility results. East-West Arctic Ocean sea ice thickness anomalies have been discussed in previous studies, with links to

Deleted: better

Deleted: accurately

Deleted: Our

large-scale atmospheric dynamics including the North Atlantic Oscillation (NAO) (Zhang et al., 2000) which could help diagnose the cause of regional biases across models. Understanding the underlying drivers of bias was considered beyond the scope of this study. The Southern Ocean reanalysis results were more mixed, and not as clearly divisible by region or longitude. More sophisticated machine learning tools, e.g. principal component analysis or self-organizing maps, could provide more insight into the dominant regional sea ice bias patterns in the Southern Ocean.

Model calibration efforts depend crucially on the research question posed. Our study focussed on plausibility methodologies and the potential benefits of ICESat-2 sea ice altimetry data for evaluating global climate model outputs of sea ice, with a brief demonstration of the impact on constraining seasonal cycles in sea ice volume across both hemispheres. We used the hemisphere-mean plausibility scores in a simple exclusion subset approach, but weighting based on these scores is an alternative option. Additional work could explore the resultant impact of our constrained/plausible subset on associated metrics including surface atmosphere-ice-ocean fluxes and sea ice freshwater fluxes within and out of the polar regions, building on previous CMIP6 studies (Keen et al., 2021; Zanowski et al., 2021). In addition, we hope that these shorter time-period mean observational data constraints can provide added benefits when combined with more commonly used plausibility metrics (e.g. sea ice sensitivity to temperature over the multi-decadal time-period) and/or recalibration approaches (e.g., Bonan et al., 2021) to provide better constrained longer-term CMIP6 sea ice predictions across metrics. Again, something we hope to explore in future work with the community.

Finally, assessments of forced sea ice-ocean models, regional models or sea ice/ocean reanalyses could benefit from similar freeboard/thickness diagnostics presented in this study. The increased emphasis on accuracy and agreement with observations would mitigate internal variability considerations and increase focus on the observational uncertainty estimates and associated issues including representation error. We chose to focus here on fully coupled climate models to explore the various additional issues introduced by the coupled model systems, e.g. the benefit of a large spread in model configurations and outputs, and the challenge of internal variability attribution and contribution. Models with sea ice embedded in the ocean (instead of the levitating assumptions typical of GCMs) may have additional motivation to assess and constrain freeboard.

#### 1050 5 Conclusions

This study provided a comprehensive evaluation of sea ice simulations from the Coupled Model Intercomparison Project Phase 6 (CMIP6) using ICESat-2 altimetry observations, in addition to sea ice area from passive microwave, introducing new insights into model plausibility and constraints. Freeboard comparisons benefit from accurate observations by satellite laser altimetry and motivate increased focus on bulk sea ice density. The short record increases the challenge in internal variability assessments, which generally contributed more than our observational uncertainty estimates to our plausibility metrics. While CMIP6 Arctic sea ice simulations showed reasonable agreement with ICESat-2 freeboard and thickness data, especially for the multi-model mean, more significant biases were present in the Southern Ocean CMIP6 models, as was largely expected

from previous studies. We adopted a similar plausibility approach at the grid-scale to highlight the regional manifestation of these model bias and potential regions of implausibility across the models. An East-West composite approach highlighted the consistent model agreement in the types of regional biases observed, often linked to key dynamical features of the ice cover. The regional Southern Ocean analysis was more mixed and could benefit from additional analysis to find consistent patterns of agreement/plausibility.

We demonstrated an example of our plausibility constrains on the seasonal cycles in both Arctic and Southern Ocean sea ice volume, with the freeboard and thickness data providing crucial additional impacts over the standard area constraints in terms of the mean monthly values and inter-model spread, highlighting the role ICESat-2 data can play in CMIP6 model evaluation and constraint.

Future research should prioritize improved uncertainty quantification and expanded assessments of associated metrics in the constrained analysis. More years of data from ICESat-2 and leveraging altimetry data from prior and on-going satellite missions, e.g. NASA's original ICESat mission and ESA's CryoSat-2, could increase the utility of the freeboard and thickness constraints presented here. The study also emphasizes the importance of addressing challenges in regional sea ice dynamics, which could form the basis of future model development assessments. With the upcoming release of CMIP7, we suggest these new assessment concepts can be utilized in tandem with more traditional assessment methods to better constrain current and future variability in sea ice and their associated climate impacts.

#### Data availability

CMIP6 data in the cloud can be accessed using the <code>intake.open\_esm\_datastore()</code> function from the intake-esm Python library. The following JSON files serve as catalogs:

- Sea ice freeboard data: https://storage.googleapis.com/cmip6/cmip6-pgf-ingestion-test/catalog/catalog.json
- All other CMIP6 variables: https://cmip6-pds.s3.amazonaws.com/pangeo-cmip6.json

The esgf-pyclient package provides a Python interface for searching and accessing datasets from ESGF, including CMIP6 data not hosted in the cloud. It allows users to query ESGF metadata, filter search results, and retrieve URLs for downloading netCDF files. CMIP6 data can be searched on the Lawrence Livermore National Laboratory (LLNL) ESGF node using esgf-pyclient and the following URL: https://esgf-node.llnl.gov/esg-search. Data can then be loaded via OPeNDAP and the Xarray Python Library.

We use both final (OSI-450-a, up to 2020, OSI SAF, 2022a) and interim (OSI-430-a, 2021 onwards, (OSI SAF, 2022b)) OSI SAF sea ice concentration data which can be accessed from the THREDDS Data Server hosted by the Norwegian Meteorological Institute and loaded via OPeNDAP with the following example URLs:

- OSI-450-a: https://thredds.met.no/thredds/catalog/osisaf/met.no/reprocessed/ice/conc 450a files/monthly/catalog.html
- OSI-430-a: https://thredds.met.no/thredds/catalog/osisaf/met.no/reprocessed/ice/conc cra files/monthly/catalog.html

ATL20 total freeboard data (we use Version 4 in this study) is hosted officially through the NSIDC at: https://nsidc.org/data/atl20/versions/4 (Petty et al., 2023a).

ICESat-2 IS2SITMOGR4 sea ice thickness data (we use Version 3 in this study) is hosted officially through the NSIDC at: https://nsidc.org/data/is2sitmogr4/versions/3 (Petty et al., 2023b).

#### 1095 Acknowledgements

Thanks to the creators and maintainers of the NASA-supported CryoCloud community for providing the AWS cloud based Jupyter Hub that was used for all of our analysis. The platform radically simplifies the process of getting spun-up on AWS resources and enables team members to work collaboratively in the same compute environment. Thanks also to members of Pangeo and the Learning the Earth with Artificial Intelligence and Physics (LEAP) center who created, maintain and also continue to update the Pangeo Analysis-Ready Cloud-Optimized CMIP6 data catalog. Thanks to the various modelling groups who have worked on developing and producing the CMIP6 runs used here, as well as the groups who produce and develop the various observational products used in this study, including members of ICESat-2 mission and the EUMETSAT Ocean and Sea Ice Satellite Application Facility (OSI SAF). Thanks also to David Bailey of NCAR for very useful discussions on the internal physics of CESM2/CICE and the calculation of ice freeboard.

#### 1105 Author contributions

AP planned the study and drafted the manuscript with <u>significant input from CC and MS. CC wrangled and analysed the data</u> and generated all the figures with input from AP/MS. MS produced the schematic and helped plan/draft the paper.

# Competing interests

The authors declare that they have no conflicts of interest.

#### 1110 Financial support

All team members were supported through a NASA ICESat-2 Science Team award (80NSSC23K1253).

Deleted: considerable

Deleted: undertook the data analysis

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

| Page 6: [1] Deleted   | Alek Petty | 5/11/25 2:54:00 PM |
|-----------------------|------------|--------------------|
|                       |            |                    |
| Page 6: [1] Deleted   | Alek Petty | 5/11/25 2:54:00 PM |
|                       |            |                    |
| Page 6: [1] Deleted   | Alek Petty | 5/11/25 2:54:00 PM |
| ,                     |            |                    |
| Page 6: [1] Deleted   | Alek Petty | 5/11/25 2:54:00 PM |
|                       |            |                    |
| Page 6: [2] Formatted | Alek Petty | 5/11/25 2:45:00 PM |
| Font: 9 pt            |            |                    |
| Page 6: [2] Formatted | Alek Petty | 5/11/25 2:45:00 PM |
| Font: 9 pt            |            |                    |
| Page 6: [3] Formatted | Alek Petty | 5/11/25 2:45:00 PM |
| Font: 9 pt            |            |                    |
| Page 6: [3] Formatted | Alek Petty | 5/11/25 2:45:00 PM |
| Font: 9 pt            |            |                    |
| Page 6: [4] Formatted | Alek Petty | 5/11/25 2:45:00 PM |
| Font: 9 pt            |            |                    |
| Page 6: [4] Formatted | Alek Petty | 5/11/25 2:45:00 PM |
| Font: 9 pt            |            |                    |
| Page 6: [5] Formatted | Alek Petty | 5/11/25 2:45:00 PM |
| Font: 9 pt            |            |                    |
| Page 6: [5] Formatted | Alek Petty | 5/11/25 2:45:00 PM |
| Font: 9 pt            |            |                    |
| Page 6: [6] Formatted | Alek Petty | 5/11/25 2:45:00 PM |
| Font: 9 pt            |            |                    |
| Page 6: [6] Formatted | Alek Petty | 5/11/25 2:45:00 PM |

l

I

1

l

1

1

| Page 6: [7] Formatted  | Alek Petty | 5/11/25 2:45:00 PM |
|------------------------|------------|--------------------|
| Font: 9 pt             |            |                    |
| Page 6: [7] Formatted  | Alek Petty | 5/11/25 2:45:00 PM |
| Font: 9 pt             |            |                    |
| Page 6: [8] Formatted  | Alek Petty | 5/11/25 2:45:00 PM |
| Font: 9 pt             |            |                    |
| Page 6: [8] Formatted  | Alek Petty | 5/11/25 2:45:00 PM |
| Font: 9 pt             |            |                    |
| Page 6: [9] Formatted  | Alek Petty | 5/11/25 3:13:00 PM |
| Superscript            |            |                    |
| Page 6: [9] Formatted  | Alek Petty | 5/11/25 3:13:00 PM |
| Superscript            |            |                    |
| Page 6: [10] Formatted | Alek Petty | 5/11/25 2:45:00 PM |
| Font: 9 pt             |            |                    |
| Page 6: [10] Formatted | Alek Petty | 5/11/25 2:45:00 PM |
| Font: 9 pt             |            |                    |
| Page 6: [10] Formatted | Alek Petty | 5/11/25 2:45:00 PM |
| Font: 9 pt             |            |                    |
| Page 6: [11] Formatted | Alek Petty | 5/11/25 2:45:00 PM |
| Font: 9 pt             |            |                    |
| Page 6: [11] Formatted | Alek Petty | 5/11/25 2:45:00 PM |
| Font: 9 pt             |            |                    |
| Page 6: [11] Formatted | Alek Petty | 5/11/25 2:45:00 PM |
| Font: 9 pt             |            |                    |
| Page 6: [12] Formatted | Alek Petty | 5/11/25 2:31:00 PM |
| Font: Not Italic       |            |                    |
| Page 6: [12] Formatted | Alek Petty | 5/11/25 2:31:00 PM |
| Font: Not Italic       |            |                    |
| Page 6: [12] Formatted | Alek Petty | 5/11/25 2:31:00 PM |

I

١

l

1

l

l

l

l

l

I

l

I

Font: Not Italic

| Page 6: [13] Deleted Alek Petty 5/11/25 2:40:00 PM | Page 6: [13] Deleted Alek Petty 5/11/25 2:40:00 PM | Page 6: [13] Deleted | Alek Petty | 5/11/25 2:40:00 PM |
|----------------------------------------------------|----------------------------------------------------|----------------------|------------|--------------------|
|                                                    | Page 6: [13] Deleted Alek Petty 5/11/25 2:40:00 PM |                      |            |                    |
| Page 6: [13] Deleted Alek Petty 5/11/25 2:40:00 PM | v                                                  |                      |            |                    |
|                                                    | Y                                                  | Page 6: [13] Deleted | Alek Petty | 5/11/25 2:40:00 PM |