# Peer review of "Constraining CMIP6 sea ice simulations with ICESat-2"

_EGUsphere, 2025_

## Author Comment (AC1)

This study conducted a comprehensive comparison with ICESat-2 data and CMIP6 sea ice model outputs. The writing and structure is general good. I have only a few minor comments here:

We sincerely thank the reviewer for taking the time to provide this constructive feedback and on our manuscript. Please see below for our responses (in blue).

Figure 1: the quality of this fugure should be improved by using vectorized format.

Yes, agreed. We will make this change.

Eqn 1: I suggest to use a table to include all model constants and variables so that we can be more clear which one is constant and which one is model outputs and which one is calculated in this study.

Yes agreed, we will add a table with model variables, constants and link this to the variables included in Table 1.

Eqn 3: use same math symbol format for h_s

Yes, agreed. We will make this change.

L188: does the sea ice mass include snow?

No, this is just the sea ice mass, snow mass is provided in a different variable.

L191: Table 1, not 2

Yes, agreed. We will make this change.

L214: what is sea area percentage?

This was a bit confusingly described, this is effectively the ocean fraction within a grid-cell. We will clarify this better in the revised manuscript.

A bit of background: The standard name of sftof is sea area percentage; when used as the land/sea mask, it is divided by 100 to give the ocean fraction within a grid cell. This is typically a value of 1 where there is ocean and 0 where there is land, which is what the regridding function from xESMF expects when providing a land/ocean mask (an ocean mask is also provided for the destination grids). xESMF treats NaN values as valid data and, without the mask, would expand those NaN values into the ocean domain as shown in this masking example: https://pangeo-xesmf.readthedocs.io/en/latest/notebooks/Masking.html. The land mask is crucial with conservative normed regridding, which accounts for the fractional overlap between source and destination grid cells. The regridding algorithm (1) finds all source cells that overlap a destination cell, (2) multiplies source values by their overlapping area fraction, and (3) sums

and normalizes to compute the destination value (e.g., if a NaN value is included in an overlap, then without a mask, the resulting destination would be NaN).

L240: Give the full name of NESOSIM if it appears for the first time

Yes, agreed. We will make this change.

L243: remove the comma after et al.

Yes, agreed. We will make this change.

L245: change vs to versus

Yes, agreed. We will make this change.

Fig 2: (a-c) is the total freeboard? Please clarify; (g-i) I think it should be sea ice concentration as the unit of colorbar is %. In the caption, please change (top) to (a, b, c), (middle) to (d, e, f), (bottom) to (g, h, i), same for other figures.

Yes, these refer to total freeboard and concentration. We will make these changes!

L266: so in the IS2SITMOGR4 dataset, there are two density approaches, a constant and a J22 parameterization?

Yes, J22 ice density and derived thickness is included in the IS2SITMOGR4 v3 dataset, but should be considered a highly experimental product variable. We will make describe this more clearly in the revised manuscript.

Fig 3: similar comments as for Fig 2

Yes, agreed. We will make this change.

Table 2: use consistent unit of freeboard through out the paper. Here is cm, but it is m in the figures

Yes, agreed. We will make this change.

Eqn 7: from this definition the plausibility index should be positive, then why there are negative values in Figure 9? and can we interprete this equation that the plausibility is big when \phi is small?

Yes there was a mistake here, we will remove the the mod term as we include the direction of the model bias associated with the plausibility index, as shown in the figures. The overall idea is that the lower the value the more plausible, with a zero value indicating perfect model agreement with observations. We will better clarify this in the revised manuscript!

Section 3.1: I suggest to put the analysis of total freeboard prior to bulk ice density, as the inversion of ice density rely on the data of freeboard.

Yes, agreed. We will make this change.

L407: Eq 4, not 5

We will make this change.

L410: Eq 5, not 6

We will make this change.

L416 and other places: use symbole ±

We will make this change.

Fig 5: use a and b instead of left and right

We will make this change.

Fig 7 and 8: have you tried the relative error instead of showing absolute errors? Also, add (a) and (b) for left and right subplots.

Thanks for the suggestion. We did have some earlier versions in this format but ultimately decided the figures were much more intuitive for readers with the absolute values included.  We will add the letter labels to the revised figure.

L545: I would suggest to use "Arctic" and "Antarctica" instead of "Arctic Ocean" and "Southern Ocean" or something different, "Southen Ocean" feels like a much broader region than sea ice actually exists.

Southern Ocean is commonly used for referring to sea ice as its resides in the ocean rather than on the continent, and both "Southern Ocean" and "Antarctic" are used regularly throughout the sea ice literature. We appreciate the suggestion, but prefer to retain original terminology.

Fig 9: why there are negative values while \phi is defined as only positive

As our response above mentioned, this was an error. We removed the mod to include a sign here related to the bias direction.

Fig 11: put the explanation of those hatchings in the caption

Yes, agreed. We will make this change.

L644: fontsizes in "there" not consistent

We will make this change.

Section 4: Here I was expecting some more discussions of the difference between model outputs and observations, and also some suggestions to the modeling community. Afterall, there are quite a bit of implausible regions for both Arctic and Antarctica for some model results.

Discussion of implausible regions was omitted from the submitted draft as we were a bit concerned with the manuscript length. However, we agree more discussion is warranted. We will include a couple of suggestions for the modelling community in the revised manuscript, especially related to the stronger regional biases.

---

## Author Comment (AC2)

Review of the manuscript egusphere-2025-766 titled *"Constraining CMIP6 sea ice simulations with ICESat-2"* by Petty et al.

This manuscript describes an evaluation of CMIP6 sea ice simulations using observations from NASA's ICESat-2, focusing on sea ice total freeboard and Arctic winter sea ice thickness—metrics not traditionally used in global climate model evaluation. The authors present a plausibility framework accounting for observational uncertainty and internal variability and explore its application in both hemispheres, ultimately suggesting the incorporation of altimetry data into model assessment pipelines and an increased focus on bulk sea ice density.

The manuscript clearly fits the journal's scope: it develops and evaluates Earth system models (here, sea ice representation in climate models), integrates observational constraints, and offers methodological innovations of interest to the modeling and remote sensing communities.

Even though I am not an expert in global climate models, I found the manuscript generally easy to read and well structured. However, I think it will benefit from minor changes, which I shall detail below, and a thorough proof-reading. I find the manuscript ready for publication after minor (albeit a rather lengthy list of) revisions.

We sincerely thank the reviewer for taking the time to provide this constructive feedback on our manuscript. Please see below for our responses (in blue).

General comments:

As is typical for a modelling study, and even more so for an intercomparison, there is an abundance of acronyms, abbreviations, and symbols. Perhaps a glossary or a table defining them could be helpful (in appendix)?

Yes, agreed. We will add a table in the revision.

As a non-expert, I think a conceptual flowchart early in the paper would guide (especially non-expert) readers through the multiple datasets, metrics, and model subsets used. I recommend adding such a figure.

Yes good idea, we will make this change.

Journal's guidelines are not completely followed, I will detail below those that caught my eye.

Thank you for catching these mistakes, and we will implement them in the revision.

Detailed comments:

Abstract: Per journal's guidelines, abbreviations need to be defined in the abstract and then

again at the first instance in the rest of the text. However, abbreviations CMIP6 and GCM are not used at all in the abstract. CMIP6 is perhaps necessary, as it appears in the title (although abbreviations should be avoided there), but there's no need for GCM. Furthermore, ICESat-2 is not explained in the abstract, but it is in the main text.

Yes, fair points. We will make these changes to the abstract.

L30ff: Throughout the manuscript, the reference to the very relevant paper Notz & SIMIP Community (2020) takes different forms (at least Community, 2020; Notz & SIMIP Community, 2020; Notz and SIMIP Community, 2020; Notz and Community, 2020). Furthermore, it appears wrong in the reference list under "Community, S" and not under Notz.

Yes, agreed. We will make references consistent throughout.

L44ff: please reword "improvements … have been suggested, suggesting improvements…"

Yes, good idea. We will make this change.

Fig. 1: I cannot find a single reference to this figure in the manuscript. Please increase the font size, it's insufficient. The only thing I can see from it are the (undefined) abbreviations in capital letters.

Yes, good point. This must have been dropped in the final edits. We will add a reference and make clear the abbreviations.

L78: This is the only occurrence of $CO_2$, better to spell it out as carbon dioxide.

We will make this change.

L96: Which spelling are you using, "(kilo)meter" or "(kilo)metre"?

We used kilometre which appears to be the correct (British/European) version for Copernicus journals.

L105 (and before): "their more limited temporal coverage", I would have expected a sentence or two about the reasons here in the introduction. Issues concerning summertime altimetry data are not mentioned before L228

Yes, this was omitted due to concerns about a lengthy introduction. We will add in an extra line in the revised manuscript to provide more detailed information.

L141: Per journal's guidelines, "data" is considered a countable noun --> CMIP6 data are

*Yes, agreed. We will make that change.*

L147: Typo? ESGF, not ESGP. Also, OPeNDAP is not defined.

*Yes this was a typo. We will make that change and spell out OPeNDAP: Open-source Project for a Network Data Access Protocol*

L185: Throughout the first half of the manuscript, the density units are formatted wrong. Per journal's guidelines, units of physical quantities must be formatted with negative exponents. Please check this also in all figures.

*Good spot. We will make that change across the manuscript and figures.*

L195: Earlier at L168 seawater density was 1024 kg m$^{-3}$, now 1026 kg m$^{-3}$. A bit confusing for the reader. Was the value 1024 kg m$^{-3}$ used anywhere in the study? What is the effect of this change?

*Good spot. 1026 should have been 1024, so we will make that correction.*

Table 1: Caption should be situated above the table, please correct also for the other tables. Variables simass and sivol are not explained like the others are.

*We will make that change across table captions and define those variables.*

L211: Here and elsewhere, are embedded URLs necessary in the text, especially when they are followed by a citation with a DOI/permalink? This is a bit more personal preference, but I think they interrupt the flow of reading.

*We will revise to include these in the data section.*

Fig. 2+: Per journal's guidelines, "only the first word is capitalized in headers (in addition to proper nouns)." Please check throughout the manuscript.

*Good spot. We will fix the capitalization across the headers.*

L261-264: Per journal's guidelines, common Latin phrases are not italicized nor hyphenated: in situ.

*Good spot. We will make that change across the manuscript.*

L267: The abbreviation EM is not defined nor used elsewhere. Furthermore, the sentence fails to acknowledge all instruments used in the derivation. I suggest "derived from coincident laser scanning, snow radar, and electromagnetic induction sounding data".

*Good spot. We will add in that suggestion.*

Fig. 3 caption: Per journal's guidelines, Figure --> Fig. when it's not starting a sentence. Please check throughout the manuscript.

Good spot. We will make that change across the manuscript.

L280: Not EASE 2.0 like before? Also, this abbreviation should have been explained earlier (L208).

Good spot. This will be revised to EASE 2.0.

L290: Rewording needed "large-scale basin-scale"?

Agreed. Will change.

L294: NASA Team data has --> have

Agreed. Will change.

L300: Roach et al. (2020)

Agreed. Will change.

L321: No need for capitalization: upward looking sonar

Agreed. Will change.

L323: an --> the AWI CS2/SMOS product, or do they have several?

Will change to add 'the'. Yes, they also have a non SMOS version.

L344: Typo? 2014 --> 2024

Yes, this was a typo. Will change.

L361: Abbreviation IS-2 not introduced. If you use such an abbreviation, do it consistently throughout the manuscript. Moreover, what is the Bessel correction?

We will make this consistent across the revised manuscript. The Bessel correction is used to account for the fact we are calculating the variance from a sample and not the entire population. We will clarify in the manuscript.

L364: Typo? SSP-2.45 --> SSP2-4.5

Yes, this was a typo. Will change.

L366: Typo? increasingly --> increasing

Yes, this was a typo. Will change.

L379ff: The sentence starting "We then…" is too complex. Please rephrase.

Agreed. Will change.

L391: Per journal's guidelines, Section --> Sect. (like you have written on the next line) when it's not starting a sentence. Please check throughout the manuscript.

Agreed. Will change throughout as suggested.

L392ff: I suggest Method 1) and Method 2) to avoid subsequent colons within a single sentence.

Yes, agreed. Will use that approach!

L395: The figure numbering in the Supplementary Information is broken, you have S4 twice. Thus, all the subsequent figure numbers are off. Please correct.

Good spot. Will update the numbering

L409: Do you mean CNRM-CM6-1-HR? In Fig. 4 that model doesn't have the black triangle marker.

Yes, this refers to CNRM-CM6-1-HR. We will update that.

Fig. 4: SI prefix for kilo is a lower case k, not K. Also in Fig. 10 and in the caption of Fig. 5.

Agreed. Will change.

L441: Fig. 5 has 16 models, as mentioned in its caption, too.

Good spot. Will change.

L446: Why is "Annual" capitalized? It appears so multiple times in the manuscript, but not always. Please check thoroughly. Furthermore, in Fig. 5 the multi-model mean freeboard looks more like 22 cm, not 25 cm. Is the figure correct (version)?

Yes, we will change to 'annual' throughout. This was updated in final analysis revisions, so we will update the text accordingly.

L448: But ACCESS-CM2 is shown in Fig. 5?

Yes we decided to show the version with ACCESS instead. We will update the text accordingly.

L480: Typo? inter-modal --> inter-model

Yes, this is a typo. Will correct.

Table 3: Does the upper case M for sea ice area stand for the SI prefix mega, i.e. $10^6$? If so, please use the scientific exponent notation, because area in "mega kilometers squared" does not read well with two prefixes.

Yes. Will update.

L539ff: Why do you want to highlight the models and metrics that are implausible? To me that's thinking backwards, wouldn't you want to highlight those that are plausible (/good/usable)? Think positive!

This is an understandable suggestion. However, we tested this and generated a figure that highlights plausible values instead,, but this made the plot much harder to interpret in our view. We think it best to keep this as is.

L542: highest --> best? Very high (and low) values are implausible, correct?

Yes, that was poor word choice. We will change the wording to 'best', as suggested.

Fig. 9: The colorbar should extend at both min and max ends, as values beyond ~7 exist, please add arrows/triangles at the ends indicating that (like in Fig. 11). Additionally, abbreviations should be explained (SIA/SIT/TFB/NH/SH/IA).

Good spot. We will extend the color scale and explain abbreviations.

L587ff: Again, does the upper case M for sea ice area stand for the SI prefix mega, i.e. $10^6$? If so, please use the scientific exponent notation, because area in "mega kilometers squared" does not read well with two prefixes. Furthermore, looking at Fig. 10, the volume changes are $10^3$, not $10^6$. Which one is correct?

Godo spot. We will use the exponent notation throughout to avoid confusion! That should have been $10^3$.

L610: Typo? Grid-scale

Yes. Will change.

L611: uncertainties … was --> were

Agreed. Will change.

L614: Fig. --> Figure as it starts a sentence

Agreed. Will change.

L622: remove second "with"?

Agreed. Will change.

L642: Fig. --> Figure as it starts a sentence

Agreed. Will change.

L694ff/Discussion: I would recommend adding subsections to improve and clarify the structure. The subsections can be short, e.g. on sea ice bulk density, uncertainty quantification, altimetry datasets, etc. L691 could be removed completely.

This is an interesting suggestion. We prefer to use bold subtitles, which will be incorporated in the revised manuscript.

L722: split infinitive, to better consider --> to consider better

Yes agreed thanks, will change.

L724: split infinitive, the ability of models to accurately capture regional internal variability --> the ability of models to capture regional internal variability accurately

Agreed. Will change.

L784ff: split infinitive, … to better constrain current and future variability in sea ice and their associated climate impacts. --> … to constrain current and future variability in sea ice and their associated climate impacts better.

Agreed. Will change.

L832ff: Is the citation correct, particularly the indicated time period?

Good spot. Will change.

L837: Wrong author

Good spot. Will change.

L933ff: No longer in press, please add DOI

Good spot. Will change.

L976ff: Accepted and published, please update

Good spot. Will change.

L981ff: Accepted and published, please update

Good spot. Will change.

L987ff: Page range or article number missing

Good spot. Will change.

L992: Incomplete citation, add journal, pages/article number, DOI

Good spot. Will change.

L994ff: Update article number

Good spot. Will change.

L1013ff: Accepted and published, please update

Good spot. Will change.

L1021ff: Update article number

Good spot. Will change.

L1025: Incomplete citation, add journal, pages/article number, DOI

Good spot. Will change.

L1027ff: This preprint was not accepted, but resubmitted as
https://doi.org/10.5194/egusphere-2024-2821

Thanks very much for pointing that out! Will change.